# Genomic population structure associated with repeated escape of *Salmonella enterica* ATCC14028s from the laboratory into nature

**Mark Achtman**[1]*, **Frederik Van den Broeck**[2,3], **Kerry K. Cooper**[4], **Philippe Lemey**[2], **Craig T. Parker**[5], **Zhemin Zhou**[1¤]*, **the ATCC14028s Study Group**[¶]

**1** Warwick Medical School, University of Warwick, Coventry, United Kingdom, **2** Department of Microbiology, Immunology and Transplantation, Rega Institute, KU Leuven-University of Leuven, Leuven, Belgium, **3** Department of Biomedical Sciences, Antwerp Institute of Tropical Medicine, Antwerp, Belgium, **4** School of Animal and Comparative Biomedical Sciences, University of Arizona, Tucson, Arizona, United States of America, **5** Produce Safety and Microbiology Research Unit, Western Regional Research Center, Agricultural Research Service, U.S. Department of Agriculture, Albany, California, United States of America

¤ Current address: Pasteurien College, Soochow University, Suzhou, China
¶ Membership of the ATCC14028s Study Group is provided in the Acknowledgments.
* m.achtman@warwick.ac.uk (MA); zhzhm1982@gmail.com (ZZ)

**Data Availability Statement:** All data are in EnteroBase, including within interactive publicly accessibly trees and spreadsheets that also include

## Abstract

*Salmonella enterica* serovar Typhimurium strain ATCC14028s is commercially available from multiple national type culture collections, and has been widely used since 1960 for quality control of growth media and experiments on fitness ("laboratory evolution"). ATCC14028s has been implicated in multiple cross-contaminations in the laboratory, and has also caused multiple laboratory infections and one known attempt at bioterrorism. According to hierarchical clustering of 3002 core gene sequences, ATCC14028s belongs to HierCC cluster HC20_373 in which most internal branch lengths are only one to three SNPs long. Many natural Typhimurium isolates from humans, domesticated animals and the environment also belong to HC20_373, and their core genomes are almost indistinguishable from those of laboratory strains. These natural isolates have infected humans in Ireland and Taiwan for decades, and are common in the British Isles as well as the Americas. The isolation history of some of the natural isolates confirms the conclusion that they do not represent recent contamination by the laboratory strain, and 10% carry plasmids or bacteriophages which have been acquired in nature by HGT from unrelated bacteria. We propose that ATCC14028s has repeatedly escaped from the laboratory environment into nature *via* laboratory accidents or infections, but the escaped micro-lineages have only a limited life span. As a result, there is a genetic gap separating HC20_373 from its closest natural relatives due to a divergence between them in the late 19th century followed by repeated extinction events of escaped HC20_373.

URLs and accession codes. An EnteroBase workspace containing all 498 strains and their genomes and metadata can be found at http://enterobase.warwick.ac.uk/a/44709. Links for public access are included in the figure legends. The nucleotide sequences and gene annotations of the genomic islands can be downloaded from figshare (https://dx.doi.org/10.6084/m9.figshare.13503081). All other data are present in Supplementary Material except for genomic sequences which can be downloaded from EnteroBase.

**Funding:** MA was supported by grant 202792/Z/16/Z from the Wellcome Trust (https://wellcome.org/). PL was supported by grant 725422-ReservoirDOCS from the European Research Council under the European Union's Horizon 2020 research and innovation program. CTP was supported by grant CRIS project 2030-42000-055-00D from the United States Department of Agriculture, Agricultural Research Service (https://www.ars.usda.gov/research/project/?accnNo=440168). The funders had no role in study design, data collection and analysis, decision to publish, or preparation of the manuscript.

**Competing interests:** The authors have declared that no competing interests exist.

## Author summary

Clades of closely related bacteria exist in nature. Individual isolates from such clades are often distinguishable by genomic sequencing because genomic sequence differences can be acquired over a few years due to neutral drift and natural selection. The evolution of laboratory strains is often largely frozen, physically due to storage conditions and genetically due to long periods of storage. Thus, laboratory strains can normally be readily distinguished from natural isolates because they show much less diversity. However, laboratory strain ATCC14028s shows modest levels of sequence diversity because it has been shipped around the world to multiple laboratories and is routinely used for analyses of laboratory evolution. Closely related natural isolates also exist, but their genetic diversity is not dramatically greater at the core genome level. Indeed, many scientists doubt that such isolates are natural, and interpret them as undetected contamination by the laboratory strain. We present data indicating that ATCC14028s has repeatedly escaped from the laboratory through inadvertent contamination of the environment, infection of technical staff and deliberate bioterrorism. The escapees survive in nature long enough that some acquire mobile genomic elements by horizontal gene transfer, but eventually they go extinct. As a result, even extensive global databases of natural isolates lack closely related isolates whose ancestors diverged from ATCC14028s within the last 100 years.

## Introduction

How extensively do *Salmonella* diversify when they escape from the laboratory into nature, and how widely do they spread? Here we compare the global genomic diversity acquired by a strain of *Salmonella enterica* over a period of 60 years within the laboratory environment with the diversity it acquired in nature after repeated escape.

*S. enterica* strain CDC 60–6516 (serovar Typhimurium) was isolated in 1960 from pooled heart and liver samples from 4-week old chickens [1]. Soon thereafter, it was stored at the American Type Culture Collection (ATCC) as ATCC® 14028 (S1 Text). During storage, rough variants with short lipopolysaccharide chains arose spontaneously [1]; the original, smooth variant is referred to as ATCC14028s to distinguish it from rough variants which were inadvertently sold under the original name [1]. ATCC14028s can be purchased from the ATCC or under different designations from the National Collection of Type Cultures in London (NCTC 12023), the DSMZ in Braunschweig (DSM 19587) or the Collection de l'Institut Pasteur in Paris (CIP 104115). The same strain is also listed in the WFCC-Mircen World Data Centre for Microorganisms (WDCM 00031) (http://refs.wdcm.org/strainid.htm?p=1), and can be purchased from providers of laboratory chemicals (e.g. Sigma-Aldrich, Lenticule Disc). Sub-cultures from these various sources are also maintained at multiple universities and microbial diagnostic facilities around the globe, and are used as quality control standards for growth of *Salmonella* on laboratory media (ISO 6579–1:20170), including ring trials of laboratory proficiency. ATCC14028s has also been used to test the effects of mutational changes on infection and disease in animal models ('laboratory evolution') [2–4].

ATCC14028s was the microbiological agent used for bioterrorism in 1984 by followers of Bhaghwan Shree Rashneesh who contaminated salad bars in multiple restaurants in Oregon with *S. enterica* in a failed attempt to influence local elections [5]. 751 human cases of salmonellosis ensued whose bacterial isolates resembled ATCC14028s by microbiological criteria [6]. Similar to the spread of *Salmonella* serovar Typhi over decades by Mr. N the milker [7] or Typhoid Mary [8], feces shed by these infected individuals into the sewage system could have

contaminated the environment and infected humans and animals. Additional serovar Typhimurium strains from several other contemporary outbreaks of salmonellosis in Oregon were also similar to ATCC14028s by microbiological tests, but those cases were not epidemiologically linked to the bioterrorism attack [5]. The genetic relationships of those natural "sib" isolates to ATCC14028s cannot be evaluated today because the microbiological tools that were available in the 1980s only allowed very coarse genetic resolution.

ATCC14028s has also infected humans after accidents in laboratories or during the disposal of biological waste. Pulsed-Field Gel Electrophoresis (PFGE) indicated that salmonellosis in a technical assistant in Regina, Saskatchewan reflected laboratory contamination with ATCC14028s [9]. However, genomic analyses of single nucleotide polymorphisms (SNPs) in those strains revealed up to nine SNPs that differentiated laboratory and natural isolates, and showed that the isolate from the laboratory infection was most similar to an ATCC14028s variant from a different laboratory. Thus, genetic micro-diversity seems to exist among stored cultures of ATCC14028s, possibly even as much as between related natural isolates. This conclusion was also supported by a phylogenetic SNP tree of 29 genomes of bacteria isolated from laboratory infections with ATCC14028s and of natural isolates from lettuce and other food products [10].

ATCC14028s is widely used across the United States to teach undergraduate students in laboratory courses about biological sciences [10]. Serovar Typhimurium with the PulseNet PFGE pattern (JPXX01.0014) which is typical of ATCC14028s [11] was isolated between August 2010 and June 2011 from 109 cases of salmonellosis in 38 states of the United States. Many of the infected individuals had participated in undergraduate lab courses, or had a history of contact with a microbiological laboratory. The same PFGE pattern was identified from 41 infections associated with undergraduate laboratories in a second national outbreak between November 2013 and May 2014 [12]. In 2017, another 24 strains with the same PFGE pattern were isolated from comparable sources in 16 states [13]. The three sets of infection dates each overlapped with the collegiate academic year, leading to the published conclusions that novice microbiology students had been infected during microbial training. However, the repeated temporal clustering is also reminiscent of outbreaks due to contaminated food, which might be a more likely cause of parallel outbreaks at the national level.

We recently sequenced 10,000 *Salmonella* genomes [14], including multiple serovar Typhimurium strains from diverse geographical and environmental sources. A number of those genomes closely resembled that of ATCC14028s, but were not obviously linked to laboratory contamination. We therefore considered an alternative explanation for the earlier observations described above, namely that ATCC14028s is widespread in the environment and routinely infects food products consumed by humans. Korves *et al*. [10] independently formulated the same hypothesis in 2016, but were not able to test it due to the scarcity at that time of available genomes from environmental isolates. However, in early 2021, EnteroBase [15] contained draft genomes and their metadata from more than 280,000 *Salmonella* isolates from diverse sources around the globe, including many from environmental sources. We therefore examined the genomic properties of ATCC14028s-like genomes in EnteroBase from both laboratory strains and diverse natural sources.

Our data demonstrate the existence of a uniform clade that has spread globally *via* human communications. Laboratory strains have been repeatedly seeded into the natural environment, and laboratory-derived and natural isolates are so similar that they cannot be reliably distinguished by core genome analyses. A small proportion of the natural isolates have acquired mobile genetic elements by horizontal gene transfer from other natural hosts, thereby confirming that they were from the environment and did not represent laboratory contaminants. However, the population structure of this clade is much more uniform than that of

other natural isolates, indicating that the release of ATCC14028s into the environment is not accompanied by long-term persistence and additional microevolution.

## Results

### Genotyping in EnteroBase

EnteroBase is well suited for exploring the biological and geographical sources of *Salmonella* because all its draft genomes have been assembled from Illumina short read sequences by a uniform and reliable pipeline, all genomes pass strict quality criteria, and EnteroBase assigns genotypes to genomes according to core genome Multilocus Sequence Typing [cgMLST] [15]. *Salmonella* cgMLST consists of a sequence-specific allelic designation for each of 3002 core genes, and a distinct sequence type integer for each unique combination of 3002 allelic integers (cgST). cgSTs are automatically assigned to multiple levels of single linkage hierarchical clustering (HierCC) based on their pairwise patristic differences in allelic contents, exclusive of missing data [15,16]. HierCC allows the ready identification and extraction of clusters of genomes at multiple hierarchical levels without requiring *ad hoc* phylogenetic trees [14–16].

Hierarchical clusters of indistinguishable cgSTs with no allelic differences are called HC0 clusters. Higher clustering levels reflect increasing maximal pairwise distances: HC5 clusters have maximal internal pairwise distances of up to five alleles, HC10 allows up to 10 alleles, *etc*. Epidemiological investigations based on HC5 or HC10 clusters have identified single source, time-delimited food-borne outbreaks or human transmission chains of *Salmonella* [17], *Escherichia* [18] or *Shigella* [19]. HierCC can also identify groups of genetically related strains independently of epidemiological investigations [14,15].

### Genomic diversity in ATCC14028s and its derivatives

In order to identify laboratory derivatives of ATCC14028s, we searched EnteroBase for strain names and other metadata containing variants of "ATCC14028", "60–6516", "NCTC 12023" or "laboratory evolution". Manual curation of the metadata and information exchanges with the laboratories responsible for the genomic sequencing led to the assignment of "ATCC14028s derivative" for multiple entries which had initially appeared to be of natural origins, and to the exclusion of two unreliable genomic assemblies from further analysis (S1 Table). We also identified and sequenced several additional strains from institutional and individual strain collections with a known history of derivation from ATCC14028s and known dates of acquisition, and included them among the ATCC14028s derivatives (S2 Table). The final dataset consisted of 156 ATCC14028s derivatives (Table 1). We stored them in Entero-Base as an Uberstrain [15] (GCF_000022165; EnteroBase barcode SAL_EA9729AA) with 155 sub-strains. Their cgSTs cluster in HierCC HC20_373, and the vast majority cluster in HC5_373 (140/155; 90%) or HC10_373 (152/155; 98%). Twenty percent of all pairs of these genomes were indistinguishable according to non-repetitive, single nucleotide polymorphisms (SNPs) in the core genome (Fig 1A). The other 80% of the pairwise comparisons formed a fat-tailed distribution with a peak at 3 SNPs and a maximum of 16 SNPs.

A fat-tailed distribution of core genomic diversity might represent the accumulation of mutations during laboratory storage and passage since 1960. Alternatively, such a distribution might reflect sequencing errors between different laboratories, or even between repeated sequencing in a single laboratory. We therefore examined the core SNP frequency distribution between genomes of *Salmonella* that had been sequenced twice. Unlike the results with lab-derived strains, 80% of the repeated sequences were indistinguishable, and only few SNPs distinguished most other pairs of repeated sequences (Fig 1A). Thus, sequencing errors are rare upon repeated sequencing, and the sequence differences between some laboratory-derived

**Table 1. Sources of 496 genomes within HierCC cluster HC20_373, including ATCC14028s derivatives (EnteroBase 1/12/2020).**

| Summary data | Laboratory-derived | Human infection | | Livestock | | | Environment | | Other |
|---|---|---|---|---|---|---|---|---|---|
| | | Laboratory | Non-Laboratory | Poultry | Swine | Cows | Plants or Food | Soil or Rivers | Animal Feed, Llama, Lamb |
| Number | 156 | 17 | 159 | 42 | 33 | 24 | 47 | 11 | 7 |
| Dates | Since 1960 | 2010–2020 | 2004–2020 | 2001–2018 | 2007–2019 | 2000–2018 | 2002–2019 | 2007–2018 | 2007–2017 |
| Geography | global | N. America France, UK | global | Americas Europe | Americas, Europe Africa | Americas Europe | Americas, Europe, Asia | North America, Europe | Americas, Europe, Asia |
| Number with transposon | 27 (17%) | 3 (18%) | 0 (<1%) | 0 (<2%) | 0 (<3%) | 0 (<4%) | 0 (<2%) | 0 (<9%) | 0 (<14%) |
| Transposon | *cat*, Tn5, Tn10 | *cat*, *tet* + Tn3, Luciferase | None | None | None | None | None | None | None |
| No. which acquired GI | 3 (2%) | 1 (6%) | 17 (11%) | 5 (12%) | 0 (<3%) | 2 (8%) | 3 (6%) | 0 (<9%) | 1 (14%) |
| GI | IncP1 plasmid, λ prophage | 1 IncI Plasmid | 14 IncI & other plasmids; 6 prophages | 1 plasmid; 4 prophages | None | Prophage | 2 plasmids; 1 prophage | None | 1 IncI1-I plasmid |
| Number which lost GI | 9 (6%) | 2 (12%) | 1 (<1%) | 0 (<2%) | 0 (<3%) | 1 (4%) | 0 (<2%) | 0 (<9%) | 1 (14%) |
| GI lost | SPI-1, SPI-11, pSV IncFII Virulence plasmid | SPI-2 | SPI-12 | None | None | Virulence plasmid | None | None | SPI-1,2,3,4,5, pSV IncFII Virulence plasmid |
| Number which lost antigen genes/Gifsy | 5 (3%) | 2 (12%) | 0 (<1%) | 2 (5%) | 0 (<3%) | 0 (<4%) | 0 (<2%) | 0 (<9%) | 1 (14%) |
| Antigen genes/ Gifsy | *fli*, Gifsy-1 glycosyltransferase | *rfb* | None | Gifsy-1 | None | None | None | None | Gifsy-1, Gifsy-3 |

NOTE: GI, any form of genomic island, including plasmids and bacteriophages.

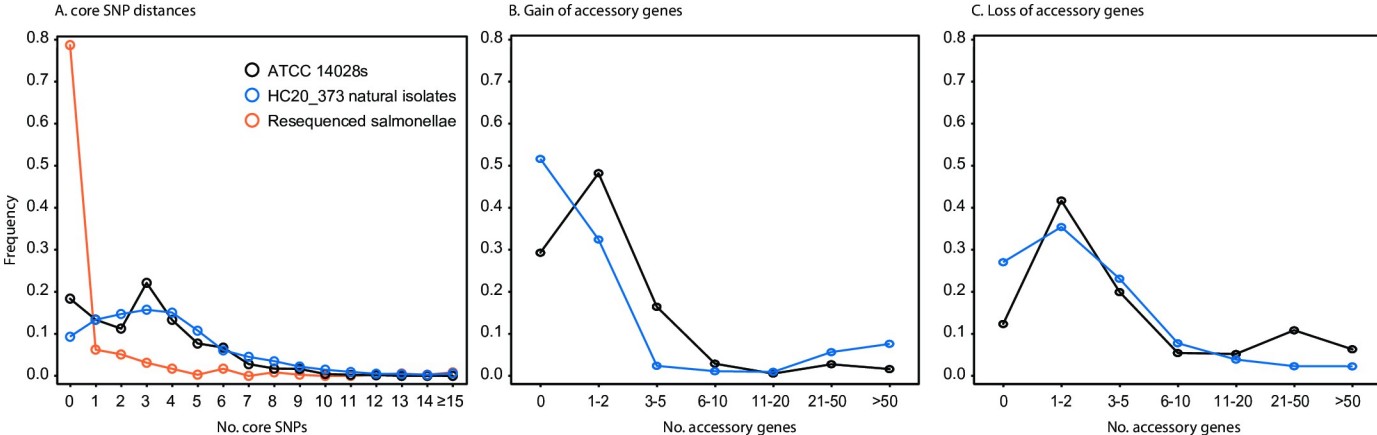

**Fig 1. Core and accessory genetic diversity within 496 HC20_373 genomes from strains stored in laboratory collections or isolated from nature.** A). Core SNP distances. Numbers of non-repetitive SNPs in an all against all comparison of pairs of genomes within each category. Maximum numbers of core SNP differences: Natural isolates, 30; ATCC 14028s, 16; resequenced genomes, 16. B). Numbers of additional accessory genes according to a pan-genome of all isolates in an all against all comparison of pairs of genomes within each category. C). Numbers of deleted accessory genes according to a pan-genome of all isolates in an all against all comparison of pairs of genomes within each category. ATCC 14028s (black): 172 strains from laboratory sources or from laboratory infections. HC20_373 natural isolates (blue): 324 strains from all other sources. Resequenced salmonellae (red; only in part A): 285 pairs of genomes that were sequenced twice for the UCCUoW 10K genomes project [14]. Additional details on the accessory genes in parts B and C can be found in S2 Fig.

variants of ATCC14028s likely reflect rare, core SNPs that have accumulated during laboratory storage and passage since 1960.

## Core genomic diversity in natural isolates of HC20_373

ATCC14028s must have existed in the environment prior to its isolation in 1960, and its close relatives (sibs) and descendants might conceivably be common among natural isolates today. ATCC14028s has also subsequently escaped from the laboratory on multiple occasion via repeated contamination of sewage systems with feces from the hundreds of people infected with ATCC14028s in 1984 during the bioterrorism attack [6] or in subsequent laboratory accidents [9,11–13]. Natural sibs and descendants of ATCC14028s would be expected to show more genetic diversity than laboratory-derived strains due to the effects of large population sizes, neutral drift and selection in nature. We tested these expectations by examining the genetic diversity of *S. enterica* HC50_147, a higher order hierarchical cluster which includes HC20_373 and other related HC20 clusters.

On Dec 1 2020, HC50_147 included 2098 entries from natural sources (Fig 2). All other HC20 clusters within HC50_147 differ from HC20_373 by at least 20 distinct alleles. We show

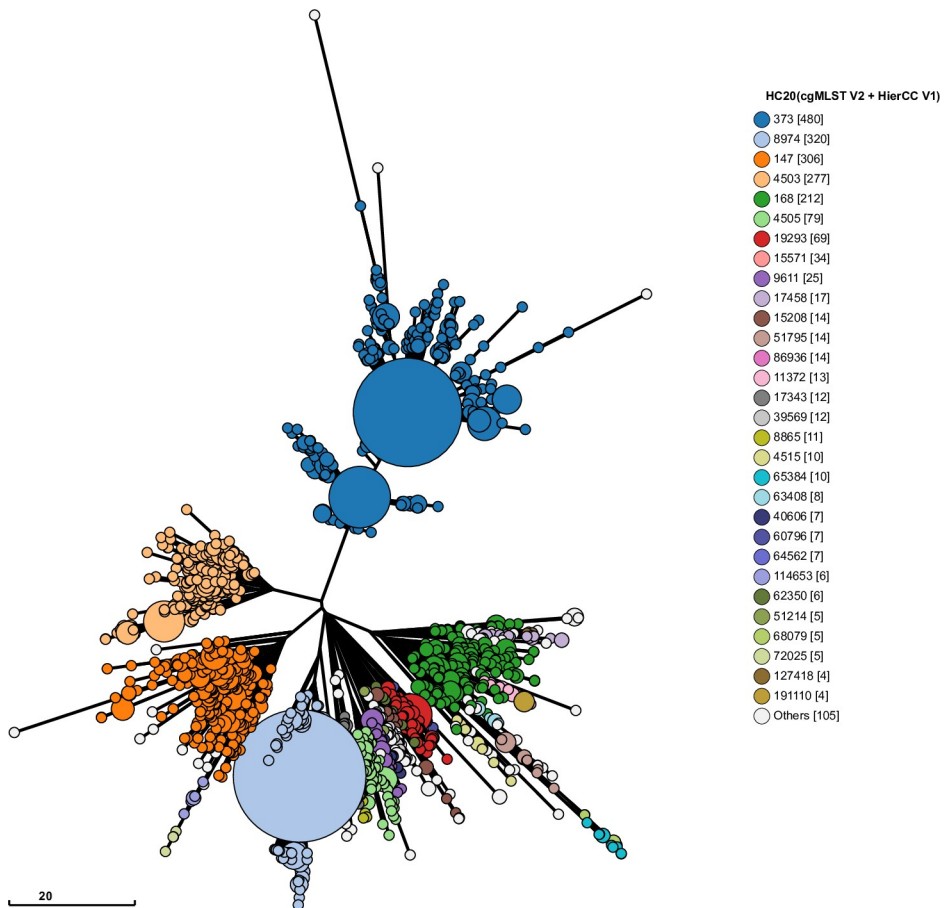

**Fig 2. Neighbor Joining tree of allelic differences in genomes in HC50_147.** Ninja NJ [64] visualization of allelic differences in the 3002 core genes of the cgMLST *Salmonella* scheme with GrapeTree [57]. At least 20 alleles differ between HC20_373 and all other HC20 clusters in the tree. The tree encompasses all 2098 HC50_147 genomes in EnteroBase on 1 December, 2020 at which time-point EnteroBase contained >270,000 *Salmonella* genomes. An interactive plot of this data can be found together with additional metadata at http://enterobase.warwick.ac.uk/ms_tree/50936.

below that the time to their most recent ancestor (tMRCA) with HC50_147 was before 1900. Any recent, natural descendants or sibs of ATCC14028s should therefore be restricted to HC20_373.

Natural isolates in HC20_373 were from diverse geographic areas and host types: half of the entries with metadata were isolated outside the U.S. (176/353) and from non-human sources (178/354). In contrast, metadata from the other HC20 clusters of HC50_147 showed that they were predominantly isolated from humans in the U.S. Only 4% (62/1580) were from outside the U.S., and only 28% (259/919) from non-human sources, demonstrating that their epidemiological properties differed from HC20_373.

We scrutinized the metadata from putatively natural isolates in HC20_373 to identify entries whose provenance might be ambiguous, and excluded several such entries from further analysis (S1 Table). We also corresponded with the individuals who had deposited their short-read sequences in EnteroBase (the ATCC14028s Study Group) in order to identify and exclude misattribution or laboratory cross-contamination among supposed natural isolates. The final set contained 17 isolates from laboratory infections and 323 from other sources, for a total of 340 natural bacterial strains (Table 1 and S3 Table). Their geographical sources were global, and these bacteria have been isolated on all continents except Antarctica since 2000 (Fig 3). Unlike the common assumption that *S. enterica* is restricted to infections of mammals and birds, recent data demonstrates that it can be readily isolated from rivers, ponds and drinking water [20–23], salt water [24,25], and reptiles [26–29]. *S. enterica* can also invade plants and survive in soil [30–32]. The HC20_373 strains in EnteroBase had been isolated from humans, livestock (poultry, swine and cows), the environment (plants, unspecified food, soil, rivers) or other sources (Table 1 and Fig 4). Most of them had been isolated from humans due to the extensive, routine sequencing since 2015 of *Salmonella* isolates for epidemiological purposes in the U.S. or the UK [33,34]. Seventeen genomes were explicitly sequenced for this project by C.P. and K.C. from isolates from plants (one strain each from cantaloupe, celery, nuts and

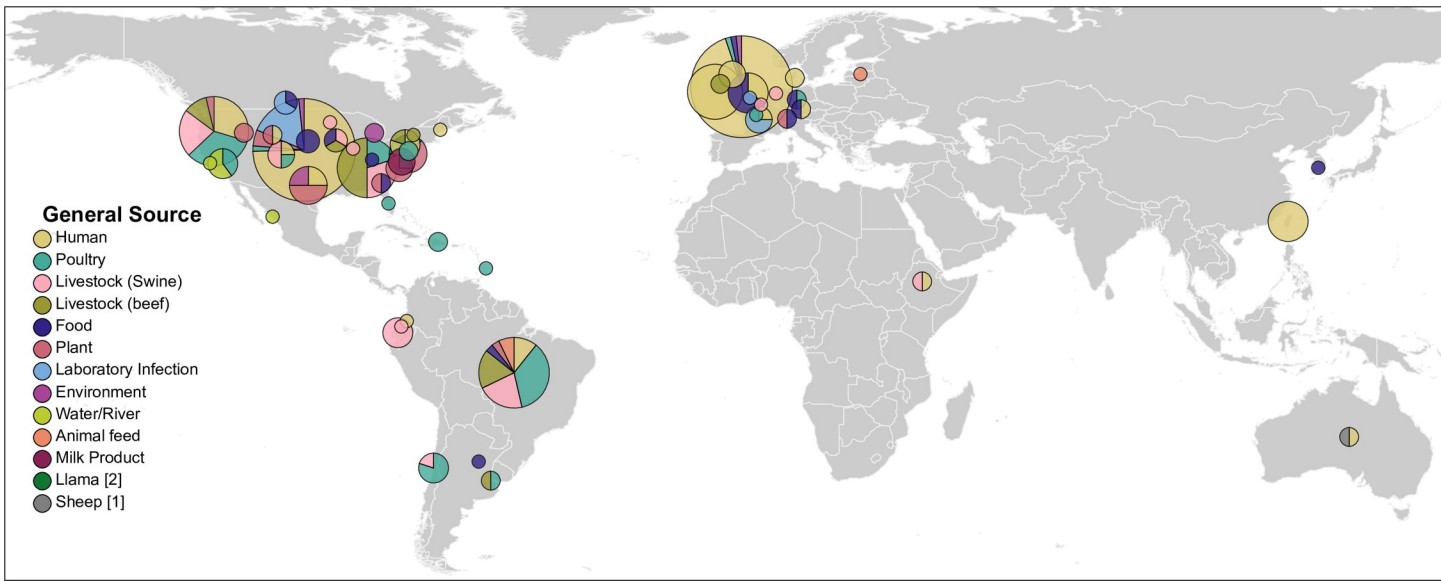

**Fig 3. Pie charts of the environmental sources of 341 natural *Salmonella* strains in HC20_373 according to geographic source.** The diameters of the pie charts are scaled to the numbers of isolates from each geographical source. Seventeen of the isolates were from laboratory infections and the others were from the sources indicated in the Key Legend (left). Map and pie charts were generated using the open-source D3.js [65] library GeographicLib [66] with the MIT/X11 license at https://github.com/d3/d3-geo/blob/main/LICENSE.

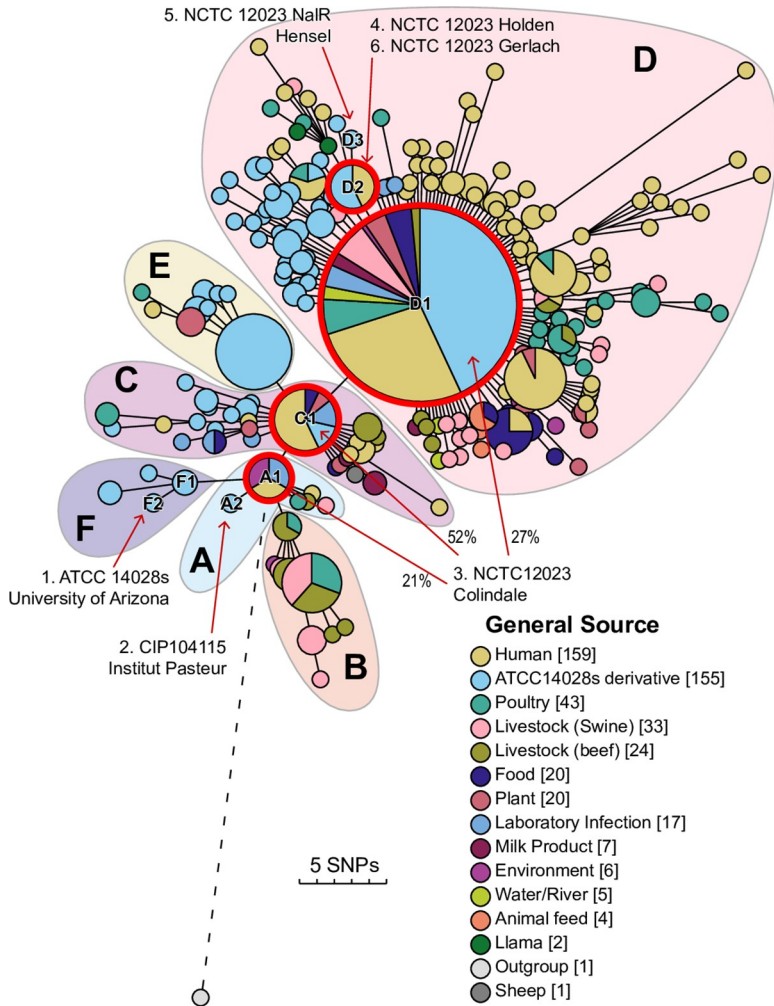

**Fig 4. Maximum likelihood (RaXML-NG [67]) phylogenetic tree of 462 non-repetitive SNPs in 496 genomes of strains in HC20_373 plus one outgroup genome (strain SAP17-7699; NCBI Accession GCF_005885875; EnteroBase barcode SAL_AB1180AA) from the related HC20 cluster, HC20_147.** The tree is presented with GrapeTree [57]. Indistinguishable genomes were collapsed into pie-chart nodes, the areas of which are proportional to the number of genomes, and whose color-coded sectors indicate their General Source (Key Legend). The phylogenetic root of HC20_373 is indicated by the branch connecting node A1 with an outgroup genome from HC20_147 (EnteroBase strain barcode SAL_AB1180AA). Other branches were rotated manually without distorting their topologies in order to cluster tips by General Source. A visual examination of the tree indicates a temporal progression from the root (A1) to the current versions of individual laboratory strains (S2 Table): 1. ATCC14028s University of Arizona (U.S.A.; node F2; 1960), 2. CIP104115 Institut Pasteur (Paris; A2; 1994), 3. NCTC 12023 Colindale (London; A1; 1987) and 4. NCTC 12023 Holden (London; D2; 1995), 5. NCTC 12023 NalR Hensel (Erlangen; D3; 1996), 6. NCTC 12023 Gerlach (Wernigerode; D2; 2003). The percentages of each of three heterozygous variant SNPs in the completed genome of NCTC 12023 (S2 Table) are indicated on lines from nodes A1, C1 and D1 to 3. NCTC12023 Colindale. This temporal progression is also in accord with an arbitrary partitioning of the tree into six sectors, A-F, consisting of apparent radial expansions of variants from central nodes (A1, C1, D1, F1, etc.). Partition D contains ATCC14028s variants from various global sources. E includes laboratory-derived mutants of ATCC14028s [68] plus their descendant nodes. F includes an early isolate of ATCC14028s (node F2) sequenced by Jarvik et al. [1]. D, E and F are separated by internal partitions A and C. Partition A was the parent of partition B, which consists of strains from natural sources. Interactive plots of the data including additional metadata can be found at http://enterobase.warwick.ac.uk/a/54094.

corn soy blend), rivers and lakes (Salinas river x 2; Salinas lake x 1) and humans (13 strains) in diverse states of the U.S. Other strains isolated in the early 2000s from global sources had been sequenced as part of the UoWUCC 10K genomes project [14], including several from humans

in Ireland and Taiwan. We thereupon sequenced 22 additional genomes (Ireland: 14; Taiwan: 8) from later isolates from human salmonellosis in those two countries which had the same PFGE patterns; almost all fell into HC20_373.

Lab infections with ATCC14028s were identified by the metadata associated with the genomes and by literature citations [9,10]. Their core genome sequences were almost identical to those of ATCC14028s, and yielded almost indistinguishable frequency distributions of pairwise, non-repetitive SNP differences. These data were accordingly combined in Fig 1A. Surprisingly, the same was true for other HC20_373 genomes from natural sources, which showed only a marginal shift towards greater pairwise differences and a maximum of 30 core SNP differences (Fig 1A).

## Comparisons of HC20_373 with other HC20 clusters

HierCC is a recent development [16], and HC20 clusters have not yet been extensively characterized. We therefore investigated the general properties of *Salmonella* HC20 clusters to determine whether HC20_373 was unusual. The numbers of HC20 clusters in the EnteroBase *Salmonella* database are inversely correlated with the numbers of genomes per cluster, dropping from >31,000 HC20 clusters each containing only a single genome down to a single cluster (HC20_2; monophasic Typhimurium) containing >16,500 genomes (Figs 5A and S1). HC20_373 belongs to a select group of 69 HC20 clusters which contained >400 genomes (Fig 5B) (0.15% of 45,964 clusters; June, 2021). HC20_373 is also highly exceptional because it demonstrated less pairwise allelic diversity than all but one of the other large HC20 clusters (Fig 5B). We chose ten HC20 clusters that had been isolated over multiple decades for deeper analysis, and compared their mean pairwise core SNP distances with HC20_373 (Table 2). Mean distances were greater for eight of these ten HC20 clusters (9.5–27.9 SNPs) than for HC20_373 (1.9) (Table 2). We also perused NJ trees of allelic diversity from these eight HC20 clusters, as well as from seven other large HC20 clusters selected arbitrarily from Fig 5B (S4 Table; S1 Text). The core genomes within these HC20 clusters were distributed over multiple, internal HC10 sub-clusters which may represent independent radiations over time from multiple central nodes (S4–S7 Figs; S4 Table; S1 Text).

These observations indicate that the population structure of HC20_373 differs from that of natural isolates. For the other HC20 clusters, environmental persistence over decades has allowed the accumulation of decentralized populations of long chains of non-repetitive SNPs. In contrast, most of the genomes in HC20_373 are concentrated in a limited number of central nodes (Fig 2). Furthermore, a Maximum Likelihood (ML) tree of non-repetitive SNPs within HC20_373 genomes (Fig 4) indicated the repeated, limited radial expansions of HC20_373 genomes from multiple, closely related founder nodes, some of which included both ATCC14028s and natural isolates. Importantly, there were no long branches in that tree, indicating only limited persistence of ATCC14028s after multiple instances of its escape to nature.

Our analyses also revealed two exceptional, large HC20 clusters. HC20_4179 (serovar Newport) and HC20_39803 (Hadar)) showed low allelic and SNP diversity despite their isolation for more than 10 years (Table 2) due to most of the genomes in each cluster being concentrated in a single central node (S8 Fig). These patterns resemble those of HC20_373 more than those of the other 15 clusters examined. Almost all HC20_4179 strains had been isolated in mid-2020 in Canada (S1 Text), and rare isolates on external branches had been isolated in previous years (S8A Fig; S1 Text). This cluster was likely associated with a large, single source onion-associated outbreak that was reported from Canada in 2020 [35]. Putative epidemiological associations for HC20_39803 were less clear, except that most genomes were from 2015–2018, when individuals in the U.S. who reared chickens at home were often infected with

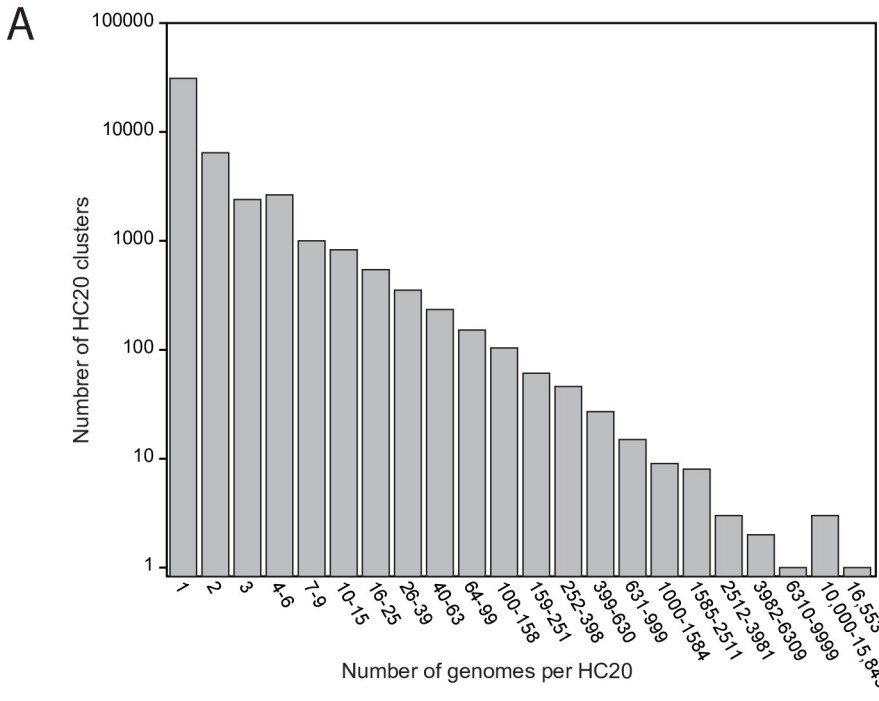

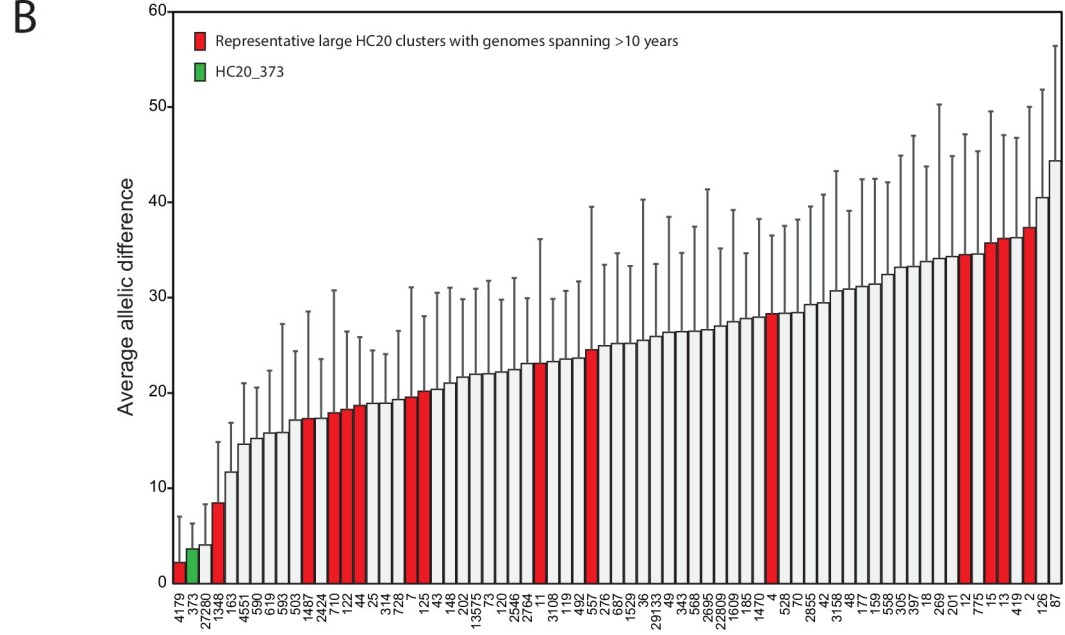

**Fig 5. Properties of HC20 clusters among >300,000 assembled genomes within the *Salmonella* database within EnteroBase (June, 2021).** A) Numbers of HC20 clusters *vs* numbers of genomes per cluster. B) Average pairwise allelic differences between genomes in HC20 clusters containing at least 400 genomes. Red indicates selected HC20 clusters whose properties are summarized in Tables 2 and S4. NJ trees of allelic distances can be found in S4–S9 Figs. HC20_373 is highlighted in green.

serovar Hadar infections [36]. Thus, the limited allelic diversity of both HC20 clusters may reflect the repeated sequencing of bacteria from two recent short-lived outbreaks of food-borne salmonellosis (S1 Text).

**Table 2. General properties of HC20_373 and other HC20 clusters.**

| HC20 cluster | Tree Number | No. genomes | Serovar | Geography | Predominant Host | Dates | Pairwise SNP Differences | | |
|---|---|---|---|---|---|---|---|---|---|
| | | | | | | | Mean | Median | Max |
| Random large HC20 clusters | | | | | | | | | |
| 10 | 45443 | 315 | Adelaide | U.S.A. | Swine (83%) | 2009–2020 | 21.5 | 16.0 | 69.0 |
| 44 | 45449 | 970 | Newport | U.S.A. | Cows (61%); Humans (23%) | 2000–2020 | 19.9 | 20.0 | 54.0 |
| 122 | 45447 | 1156 | Newport | U.S.A. | Human (55%); Environment (41%) | 1998–2020 | 25.0 | 23.0 | 65.0 |
| 125 | 45450 | 576 | Bertha | U.S.A. | Chickens (51%); Humans (29%) | 1994–2020 | 27.9 | 27.0 | 73.0 |
| 557 | 45446 | 849 | Muenchen | U.S.A. | Chickens (79%); Swine (7%) | 1988–2020 | 19.4 | 13.0 | 71.0 |
| 710 | 45445 | 420 | Johannesburg | U.S.A. | Swine (82%) | 2003–2020 | 17.3 | 12.0 | 59.0 |
| 1348 | 45423 | 844 | Enteritidis PT8 | UK | Human (95%); Feeder Mice[1] | 1972–2020 | 9.5 | 7.0 | 59.0 |
| 1487 | 45424 | 359 | Typhimurium | England | Human (97%) | 2006–2020 | 19.7 | 15.0 | 57.0 |
| Outbreaks | | | | | | | | | |
| 4179 | 60146 | 581 | Newport | North America | Human (97%) | 2010–2021 | 3.2 | 1.0 | 56.0 |
| 39803 | 45421 | 222 | Hadar | U.S.A. | Human (95%) | 2000–2018 | 3.0 | 3.0 | 17.0 |
| Lab strains | | | | | | | | | |
| 373 | 60131 | 496 | ATCC14028s | global | Laboratory and Nature | 2000–2020 | 1.9 | 1.0 | 15.0 |
| 5519 | 60183 | 46 | SL1344 | Europe | Laboratory (89%) | 2002–2018 | 4.6 | 5 | 32 |
| 20633 | 60214 | 50 | NCTC 7832 | UK, Ireland | Laboratory? | 2006–2021 | 2.6 | 2 | 11 |

NOTE: Tree number: Tree ID in EnteroBase of Ninja NJ trees of each cluster.

[1]Information from Marie Chattaway, PHE, UK has confirmed that HC20_1348 corresponds to the cluster of Enteritidis PT8 isolates that contaminated frozen feeder mice for pet reptiles [27,63].

We also investigated HC20 clusters of 50 genomes each that were associated with two other laboratory strains of *S. enterica*: serovar Typhimurium SL1344 (HC20_5519) and serovar Nottingham ATCC 7832 (HC20_20633) (Table 2 and S1 Text). Each cluster showed comparable mean SNP diversities to HC20_373 (Table 2). However, their NJ trees were somewhat stragglier (S9 Fig).

Most HC20_5519 genomes were from laboratory derivatives of SL1344 according to the accompanying metadata. If SL1344 has escaped into nature, infections of humans or domesticated animals have been very rare. However, the metadata for HC20_20633 genomes indicated that they were natural isolates. Furthermore, HC20_20633 contained the same genomes as HC5_20633, indicating that there is even less diversification within these bacteria than for the other laboratory strains.

ATCC 14028s is used for quality control of growth of *Salmonella* growth media in many countries, but in the British Isles (UK and Ireland), strain NCTC 7832 (serovar Nottingham; HC5_20633) is recommended for this purpose. Correspondingly, HC20_373 was isolated globally but 49/50 HC5_20633 genomes were from the British Isles. Based on their metadata, seven HC5_20633 strains had been isolated from humans and 26 others from other sources. These genomes might represent repeated escape from the laboratory in the British Isles. Alternatively, many of those genomes might represent repeated laboratory cross-contamination with the organism being used for routine QC of growth media during attempts to cultivated natural isolates.

## Evolutionary history of ATCC14028s

An ML SNP tree of HC20_373 (Fig 4) provides further insights on the evolutionary history of ATCC14028s, and further support for repeated escape from the laboratory. The root of this tree is defined as node A1 by the branch to an outgroup genome in an independent HC20

cluster within HC50. Based on this rooting, and other temporal reconstructions (S1 Text, S2 Table), other nodes within the ML tree were assigned to 6 internal partitions (A-F), each radiating from a founder node (A1, B1, etc.). A1 was the parent of other nodes in partition A as well as the founder nodes for B, F and C. Similarly, C1 was the parent for the other nodes in partition C and the founder nodes of D and E.

The partitions in the ML tree are also supported by the following dated history of ATCC14028s acquisitions. Root node A1 contained isolates from laboratory infections as well as natural strains isolated from humans and the environment. It differs by two core SNPs from node A2, which contains CIP 104115, a version of ATCC14028s which was acquired by Institut Pasteur, Paris in 1994. A1 differs from node F1 by three core SNPs, and F1 differs from F2 by a fourth SNP. F2 includes the complete genome of an early subculture of ATCC14028s that was sequenced at University of Arizona [1]. A1 differs from C1 by one core SNP, and C1 precedes D1 by two other SNPs. NCTC 12023 is the NCTC designation for a sub-culture of ATCC14028s which they acquired in 1987, freeze-dried and stored as a frozen stock. A complete genome of NCTC 12023 from a subculture of that stock was heterozygous for the three SNPs distinguishing A1, C1 and D1, with some reads encoding the D1 variant and others encoding the variant associated with A1 or C1 (Fig 4 and S2 Table). The heterozygosity within this culture demonstrates that all three SNPs had been gained by ATCC14028s by 1987 when it arrived in London. NCTC 12023 was acquired by David Holden in 1995 (node D2, one SNP further), and passed on to Michael Hensel who brought it to Erlangen, Germany, in 1996 where a NalR *gyrA* mutant was selected (node D3). The original Erlangen version (node D2) was sub-cultured in 2003 and accompanied his associate Roman Gerlach to Wernigerode, Germany, in 2005 (S2 Table and Fig 4). This documented history of subcultures and resulting genome sequences confirms the outgroup-based directionality in Fig 4.

Nodes C1, D1 and D2 included numerous natural isolates in addition to ATCC14028s derivatives, indicating that the core genomes of those natural isolates are indistinguishable from laboratory derivatives of ATCC14028s. Other genomes from diverse sources were widely distributed throughout HC20_373: human isolates are present in partitions A, C, D and E, livestock isolates in partitions B, C, D and E, and environmental/plant isolates in all partitions except F. However, the smaller nodes containing natural isolates largely represent a limited set of radial expansions from the central nodes containing ATCC14028s derivatives, indicating that most natural isolates are direct descendants of the laboratory strains. Similar results were obtained with minimal spanning trees or neighbor joining trees based on differences in core genome MLST alleles. Thus, natural isolates within HC20_373 represent minimal changes from sequence variants which have arisen in the laboratory, and do not seem to have undergone extensive adaptation or neutral drift.

## Are "natural" isolates truly natural?

An alternative explanation for the great similarity between natural isolates of HC20_373 and ATCC14028s might have been that the natural isolates were not truly natural. Except within the British Isles, ATCC14028s is routinely used for quality control of *Salmonella* growth media, and cross-contamination between cultures is particularly common when working with *Salmonella* unless automated microbiology is used [37]. Furthermore, the frequency of HC20_373 strains was relatively low among all *S. enterica* genomes (Table 3). Thus, similar to NCTC 7832, many "natural isolates" might simply represent occasional cross-contamination from laboratory stocks of ATCC14028s. Our manual curation identified 11 supposed isolates from vegetables in South Africa which were subsequently attributed to laboratory contamination by their source laboratories (S1 Table; excluded from further analysis in this publication)

**Table 3. Proportions of natural HC20_373 isolates in individual countries.**

| Country | Criterion for further testing | Dates | Source | Number (percentage of all isolates) |
|---|---|---|---|---|
| Ireland (Republic)[a] | B/15 (PFGE) | 2004–2020 | Human | 17/1911 (0.7%) |
| Taiwan | STX.0170 (PFGE) | 2005–2017 | Human | 9/33,851 (0.03%) |
| Denmark | (MLVA)[b] | 2007–2009 | Human | 2/8000 (0.05%) |
| Ecuador[c] | HC20_3 73 | 2018–2019 | Poultry, Swine, others | 6/896 (0.67%) |
| France | HC20_373 | 2014–2021 | Human, others | 6/22,333 (0.03%) |
| UK | HC20_373 | 2014–2020 | Human | 74/51,853 (0.14%) |
| United States (FDA) | HC20_373 | 2007–2018 | All isolates | 80/23,473 (0.34%) |

[a]Based on data within EnteroBase for all *S. enterica* that were isolated in Ireland between 1997 and 2018, HC20_373 strains consisted 0.6% (17/2799) of *S. enterica* that were isolated from humans and 0.1% (2/1822) of isolates from other sources (pig and chicken dust in Northern Ireland).

[b]MLVA patterns were only determined for Typhimurium isolates.

[c]lymph nodes and carcasses in slaughterhouses. Most isolates were serovar Infantis.

and two laboratory derivatives of ATCC14028s which had been assigned a distinct strain designation. Of four HC20_373 isolates from human infections in France, enquiries to the responsible epidemiologists revealed that one was from a laboratory infection and two others were from a mixed *Salmonella* infection of a truck driver associated with contamination during the transport of biological waste. All three genomes were relabeled in EnteroBase as laboratory infections. However, other supposed natural isolates are unlikely to have arisen by contamination with ATCC14028s. For example, the fourth human isolate from France was from a baby with no known contacts to a laboratory or biological waste, and two veterinary isolates from France were from chicken and pork, and are also considered to represent natural isolates.

Numerous other observations also support the interpretation that many HC20_373 isolates were of natural origin (Table 3). HC20_373 genomes were isolated 17 times from human infections in the island of Ireland, and twice from non-human sources (Table 3). In contrast, all eight strains isolated in Ireland of HC5_20633 (serovar Nottingham, includes NCTC 3782 which is used for QC of growth media in the British Isles) were exclusively from non-human sources (S9A Fig). England, Scotland and Wales were the source of 74 "natural" strains of HC20_373 between 2007 and 2020 (human infections: 65; non-human sources: 9), and of 25 strains of HC5_20633 (human infections: 7; non-human sources: 18). Thus, HC20_373 was isolated almost exclusively from human infections in the British Isles while HC5_20633 was isolated predominantly from other sources.

Six HC20_373 strains were isolated from swine during the course of veterinary diagnostics in Ecuador, but quality control in that laboratory depends on a monophasic Typhimurium strain rather than ATCC14028s (diphasic Typhimurium). Taiwan was the source of nine other human isolates between 2005–2017 but they also do not use ATCC14028s for routine quality control. Several other "natural" isolates were from reference laboratories in Germany, Denmark or France; each of these confirmed that their strain collections include ATCC14028s but claimed that laboratory mix-ups were extremely unlikely because microbiology was being performed with automated procedures and ATCC14028s was not in general use. Laboratory mix-ups are also unlikely for four genomes from three laboratories whose strains were isolated from the Salinas river in California and a fifth genome from a river in Sinaloa, Mexico.

## Gene gain and loss in the accessory genome

ATCC14028s is used extensively for experimental evolution, including the use of transposon mutagenesis to inactivate genes in order to elucidate their relevance to virulence. Some of the

bacterial colonies isolated after such treatments have acquired individual genes or short DNA stretches, or suffered deletions of DNA between repetitive insertion stretches. In contrast, natural isolates often exist in mixed microbial biofilms that undergo horizontal gene transfer (HGT) of plasmids, bacteriophages (including lysogenic prophages) and/or other large genomic islands, and some natural isolates could have acquired such mobile elements. We therefore compared the accessory genome of supposed natural isolates with that of ATCC14028s laboratory derivatives. To this end, a pan-genome (S5A Table) was calculated from all 496 HC20_373 genomes in this study. After excluding contaminating DNAs (S5B Table), that pan-genome consisted of 4603 genes of ≥300 bp length (S5C Table), and 98 individual genomes had gained or lost individual accessory genes or clusters of genes (S5D Table). These genetic events were assigned to 63 distinct inDel types: lab mutations with transposons (33 genomes, 13 types), deletions (30 genomes, 23 types), the acquisition of plasmids (21 genomes, 20 types) and/or lysogenization by prophages (14 genomes, 7 types) (S4C and S4D Table). The frequencies of gene gain and loss events are summarized in Fig 1B and 1C, respectively. The gene contents of the genomic islands are illustrated in S2 Fig, and the inDel types are summarized by source in Table 1 and S6 Table.

The results largely confirmed our expectations: 30/172 (17%) ATCC14028s derivatives or strains from laboratory infections had acquired typical transposons that are used for genetic manipulations while only 3 (1.7%) had acquired a natural plasmid or a prophage (Table 1). Only few accessory genes were acquired in the laboratory (Fig 1B), and almost all gene gain events were associated with laboratory transposons (S5C Table). In contrast, none (<0.3%) of the 324 natural strains from HC20_373 contained a laboratory transposon but 28 (9%) of them have acquired either a plasmid or a prophage, or both (Table 1 and Fig 1B). The best BLAST hits for almost all of these acquisitions were for mobile elements in genomes of *Escherichia coli* or unrelated *S. enterica* strains (S5D Table), consistent with an acquisition during mixed infections of humans, poultry, cows, and plants. In contrast, the loss of genomic stretches encompassing multiple genes was very rare in the natural isolates and more frequent among the laboratory derivatives (Fig 1C, Table 1 and S1 Text).

## Temporal signals for microevolution of core SNP diversity

We investigated whether statistically significant temporal signals were present in the core-genomes of the HC20 clusters in Table 2. Non-repetitive SNPs from genomes with known dates of isolation were plotted against time of isolation with TempEst [38], and the strength of temporal signals from unexceptional genomes in those root-to-tip plots was tested with generalized stepping-stone sampling (GSS) within a Bayesian framework [39]. To this end, we calculated marginal likelihoods for both strict and relaxed molecular clock models against the real (heterochronous) temporal data and against artificial (isochronous) data in which a uniform date of isolation was assumed for all genomic sequences (Table 4). Strong temporal signals result in a Log Bayes Factor of at least 5 for heterochronous *vs* isochronous analyses, and the likeliest evolutionary model (strict *vs* relaxed) has the largest Bayes Factor [39].

HC20_5519 and HC20_39803 did not yield strong support for a temporal signal, and they were not investigated further. All other eleven HC20 clusters yielded strong temporal signals for a strict clock (BF1 = 6–894), and six of them yielded even greater Bayes Factors for a relaxed clock (BF3 = 23–176), which then became the preferred evolutionary model for those HC20 clusters (Table 5). We also calculated the molecular clock rate and tMRCA for each HC20_cluster with the preferred evolutionary model using BEAST analyses [40] (Table 5).

When analyzed on its own, HC20_373 has an estimated tMRCA of 1971 (95% HPD 1950–1987), with a relaxed clock rate of $8.59 \times 10^{-8}$ SNPs per nucleotide per year (Table 5). In a

**Table 4. Statistically significant temporal signals within non-repetitive core genome SNPs in HC20_373 and other HC20 clusters.**

| HC20 cluster | No. genomes | Serovar | Dates | Marginal likelihood | | | | Log Bayes Factor | | |
|---|---|---|---|---|---|---|---|---|---|---|
| | | | | Contemp Strict | Contemp Relaxed | Strict | Relaxed | BF1 | BF2 | BF3 |
| 10 | 256 | Adelaide | 2009–2020 | -6300296 | -6300266 | -6300028 | -6300027 | 268 | 239 | 1 |
| 44 | 807 | Newport | 2000–2020 | -6741201 | -6741075 | -6740307 | -6740164 | 894 | 911 | 143 |
| 122 | 805 | Newport | 1998–2020 | -6942726 | -6942748 | -6942657 | -6942593 | 69 | 155 | 64 |
| 125 | 445 | Bertha | 1994–2020 | -6788539 | -6788497 | -6788366 | -6788190 | 173 | 307 | 176 |
| 557 | 749 | Muenchen | 1988–2020 | -6721331 | -6721275 | -6720904 | -6720738 | 427 | 537 | 166 |
| 710 | 405 | Johannesburg | 2003–2020 | -6169794 | -6169764 | -6169324 | -6169329 | 470 | 435 | -4 |
| 1348 | 781 | Enteritidis PT8 | 1972–2020 | -6845700 | -6845709 | -6845096 | -6844990 | 604 | 719 | 106 |
| 1487 | 341 | Typhimurium | 2006–2020 | -6787811 | -6787819 | -6787671 | -6787677 | 140 | 142 | -6 |
| 4179 | 377 | Newport | 2010–2021 | -6680482 | -6680478 | -6680401 | -6680406 | 82 | 72 | -6 |
| 373 | 419 | ATCC14028s | 2000–2020 | -6706284 | -6706329 | -6706189 | -6706166 | 95 | 163 | 23 |
| 20633 | 44 | NCTC 7832 | 2006–2021 | -6375734 | -6375742 | -6375728 | -6375732 | 6 | 10 | -4 |

NOTE: Data are as in Table 2, except that the HC20_5519 and HC20_39803 were excluded because they did not possess statistically significant evidence for temporal signals. Furthermore, only genomes were included with metadata for calendar year or isolation and which were not outliers in an analysis with TempEst [38].
Marginal likelihoods are abbreviated as Contemp Strict: strict clock with isochronous dates; Contemp Relaxed: relaxed clock with isochronous dates; Strict: strict molecular clock with heterochronous dates; Relaxed: relaxed molecular clock with heterochronous dates.
BF1 (Log Bayes Factor 1) is Strict vs Contemp Strict. BF2 is Relaxed vs Contemp Relaxed. BF3 is Relaxed vs Strict.

separate analysis, we also calculated a Maximum Clade Credibility (MCC) tree from HC20_373 plus an outgroup genome (Fig 6). That tree estimated the tMRCA of HC20_373 as 1983, and a branching date from the outgroup genome of 1890 (95% HPD 1858–1927) (Fig 6; EnteroBase genome barcodes in S3 Fig). Thus, the tMRCA of HC20_373 is approximately 10–20 years more recent than the date of isolation of CDC 60–6516, the ancestor of ATCC14028s, and 80–90 years after HC20_373 separated from other clusters in HC50_147. These estimates

**Table 5. Clock rate and tMRCA of eleven HC20 clusters.**

| HC20 Cluster | Preferred Model | Clock rate (95% HPD) | tMRCA (95% HPD) |
|---|---|---|---|
| 10 | Strict | $4.29 \times 10^{-7}$ ($3.78 \times 10^{-7}$,$4.80 \times 10^{-7}$) | 2005 (2003,2007) |
| 44 | Relaxed | $2.88 \times 10^{-7}$ ($2.57 \times 10^{-7}$,$3.18 \times 10^{-7}$) | 1986 (1975,1994) |
| 122 | Relaxed | $1.28 \times 10^{-7}$ ($1.04 \times 10^{-7}$,$1.52 \times 10^{-7}$) | 1969 (1960,1979) |
| 125 | Relaxed | $3.73 \times 10^{-7}$ ($3.03 \times 10^{-7}$,$4.43 \times 10^{-7}$) | 1980 (1968,1990) |
| 557 | Relaxed | $3.75 \times 10^{-7}$ ($3.23 \times 10^{-7}$,$4.30 \times 10^{-7}$) | 1980 (1972,1987) |
| 710 | Strict | $3.63 \times 10^{-7}$ ($3.22 \times 10^{-7}$,$4.02 \times 10^{-7}$) | 1998 (1995,2000) |
| 1348 | Relaxed | $2.53 \times 10^{-7}$ ($2.23 \times 10^{-7}$,$2.82 \times 10^{-7}$) | 1953 (1938,1967) |
| 1487 | Strict | $1.23 \times 10^{-7}$ ($1.06 \times 10^{-7}$,$1.42 \times 10^{-7}$) | 1971 (1960,1981) |
| 4179 | Strict | $2.36 \times 10^{-7}$ ($1.44 \times 10^{-7}$,$3.43 \times 10^{-7}$) | 2000 (1994,2007) |
| 373 | Relaxed | $8.59 \times 10^{-8}$ ($6.16 \times 10^{-8}$,$1.13 \times 10^{-7}$) | 1971 (1950,1987) |
| 20633 | Strict | $1.15 \times 10^{-7}$ ($5.69 \times 10^{-8}$,$1.78 \times 10^{-7}$) | 2004 (2001,2006) |

NOTE: The table summarizes Molecular clock rates and tMRCA for eleven HC20 clusters whose general properties and statistical support for a temporal signal are presented in Table 4. HC20_20633 contains the same genomes as HC5_20633.
Clock rates are in SNPs per nucleotide per year. The clock rates and tMRCAs were calculated as the median from 5400 unrooted trees from three independent chains, and can differ slightly from the results from rooted, single trees, such as in S3 Fig.

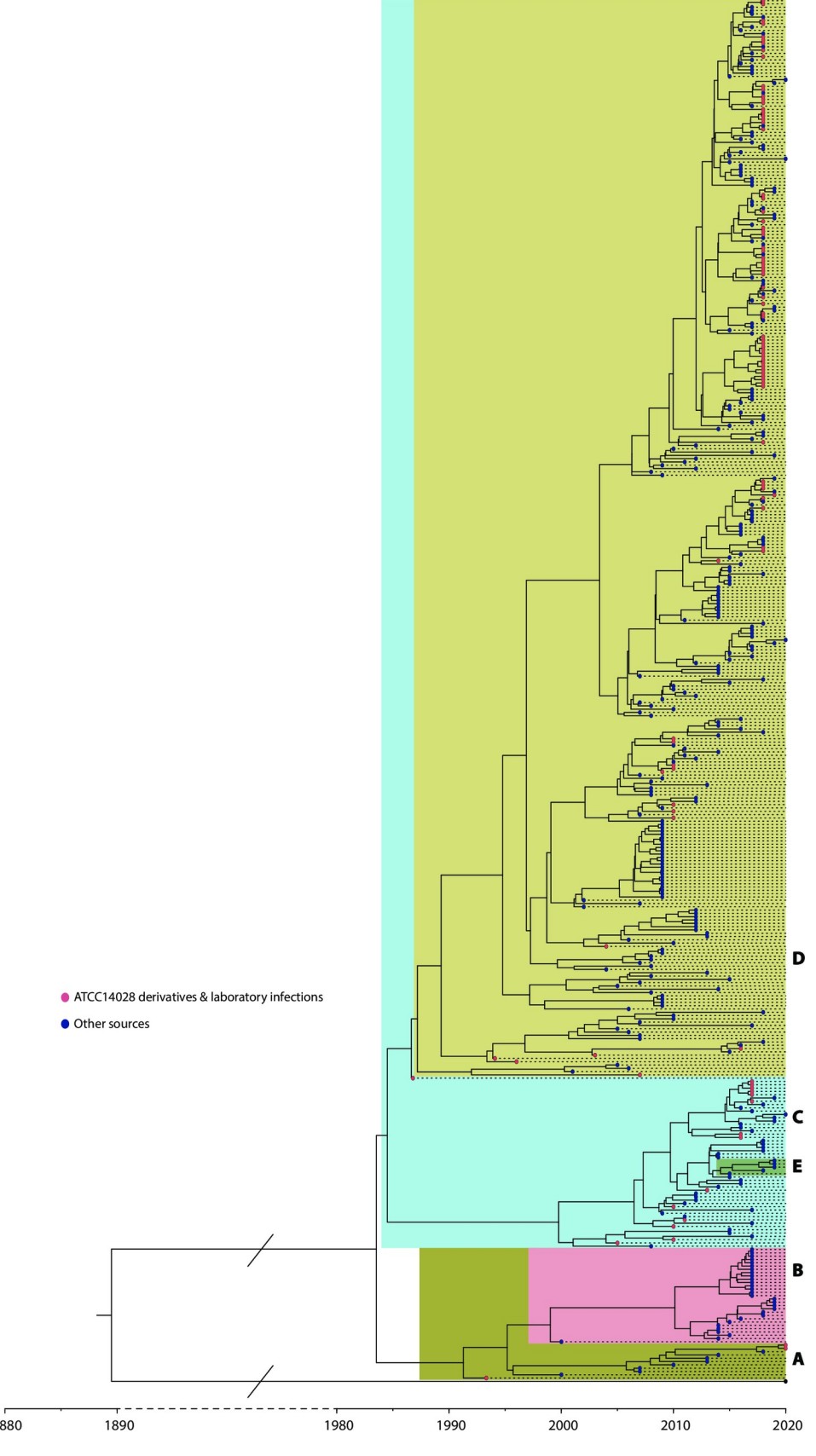

**Fig 6. Bayesian BEAST [40] temporal dating of ATCC14028s and its natural derivatives.** The figure depicts a tip-dated tree of 419 genomes from HC20_373 whose metadata included a tip date and which were not outliers in preliminary analyses (Table 4), rooted with the same outgroup genome from HC20_147 described in Fig 4. The topology of partition clustering is consistent with that of an ML tree (Fig 4), and was emphasized visually by rearranging branches manually such that they grouped by partition without changing their length or phylogenetic relationships. Partitions A-E from Fig 4 are indicated at the right and marked by distinctive blocks of colors within the tree. Partition F is lacking because no dates of isolation were available for any genomes from that partition. Tips are color-coded by Genome Source, as indicated in the Key legend. The scale at the bottom indicates the most probable calculated dates for tips and internal branch-points, but is broken between 1890 and 1980 to save space.

were obtained using a relaxed molecular clock, and prior values for Bayesian tip-dating that were calculated from an exponential distribution for the dates of acquisition of genome samples 2–4 as indicated in S2 Table. Furthermore, the MCC tree root of HC20_373 was fixed at founder node A1 (Fig 4). However, the dating estimates appear to be robust to these choices because similar tMRCA dates were estimated when samples 2–4 were excluded from analysis and no fixed rooting was imposed, or with a strict clock rate.

The other HC20 clusters yielded estimated substitution clock rates that were 1.3fold to 5.0fold faster ($1.15 \times 10^{-7}$ to $4.29 \times 10^{-7}$; Table 5), and comparable clock rates have been calculated elsewhere for several other natural clades of *S. enterica* (S7 Table). The differences in clock rates between HC20_373 and other clades do not seem to depend on the temporal depth of the data sets: the tMRCA estimates for the 10 other HC20 clusters ranged from 1953–2005 (Table 5), straddling the estimates for HC20_373.

## Discussion

### Core genome diversity in ATCC14028s

Repeated Illumina short-read sequencing of a bacterial strain is thought to usually yield the same draft genomic sequence [41,42]. Our data confirm this conclusion (Fig 1A), and we only rarely found nucleotide differences in non-repetitive regions after repeated Illumina sequencing. However, pairs of genomes sequenced from different isolates of ATCC14028s routinely differ by three to fifteen core SNPs (Fig 1A), supporting the conclusion by Jarvik *et al.* [1] that ATCC14028s has undergone microevolution in the laboratory. Further support is provided by sequential acquisition of single nucleotide variants by descendants of ATCC14028s after deposition of that strain in national culture collections in Paris (1994) and London (1987) followed by travels within London and on to Germany in the mid-1990s (Fig 4, S2 Table).

ATCC14028s was initially maintained in stab cultures for decades, and limited genomic micro-diversity would not be surprising after repeated sub-cultures and long storage of a laboratory strain. Sequence differences also accumulated during storage of LT2, another laboratory strain of *S. enterica* serovar Typhimurium that was used for genetical research and which was stored in stab cultures since 1948. Multiple archival strains of LT2 had accumulated deletions and genomic rearrangements [43], changed their phage type due to loss or inactivation of prophages [44] and/or changed their biotype [45]. However, unlike LT2, ATCC14028s is available for purchase from multiple type culture collections around the globe, and has been extensively distributed by diagnostic or research laboratories for research and ring trials. For example, NCTC sold 125 ampoules of NCTC 12023 during 2016–2021, and CIP sold 46 ampoules of CIP 104115 during 2015–2021. The strain is also commercially available from Sigma-Aldrich.

ATCC14028s genomes in the public domain include sequences from variants that were re-isolated after laboratory experiments on natural selection, including long-term exposure to antibiotics or other forms of stress. Other variants were sequenced after experimental mutagenesis with transposon vectors and/or passage through experimental animals. Many

microbiologists streak a bacterial culture to single colonies before storage, which results in an absolute bottleneck. Such bottlenecks will immediately fix any rare mutations that happen to have segregated to that single colony, and remove all others that are present at low frequency in the population. All these phenomena may well have contributed to the microevolution of ATCC 14028s and its descendants that is described here. Examples of that microevolution are very apparent in a phylogenetic tree of core genome diversity within HC20_373 (Fig 4), which is dominated by six partitions each resembling a radial expansion from a founding node. ATCC14028s genomes were found in five of the six radiations.

ATCC14028s was originally isolated in 1960 from infected chickens in the U.S. [1]. We expected to find descendants and sibs of that original source in modern domesticated animals and samples from the environment. However, the HC20_373 cluster is unusually compact, and largely consists of nodes containing multiple, indistinguishable core genomes which are joined by short branches (Fig 2). Other HC20 clusters exhibit long branches (S4–S8 Figs) and more extensive microdiversity (Table 2). Similar features apply to the neighboring HC20 clusters within HC50_147 (Fig 2). The internal phylogenetic structure of HC20_373 may largely track the core genome microdiversity accumulated in the laboratory after international transmissions by courier or mail, and only very limited additional core genomic diversity seems to have accumulated in nature (Fig 4). These observations also suggest that all the natural isolates within HC20_373 described here represent the descendants of repeated escape from the laboratory coupled with repeated subsequent extinction of those microclades (Fig 7). However, we cannot definitively exclude the possibility that the limited diversity of HC20_373 was caused by other mechanisms that reduce the rate of microevolution in nature.

## Escape of pathogens from laboratories

Laboratory accidents have resulted in multiple infections with microbial pathogens since the beginnings of microbiology in the late 19th century, in some cases with lethal consequences [46]. Escape from the laboratory has resulted in short transmission chains of smallpox [47],

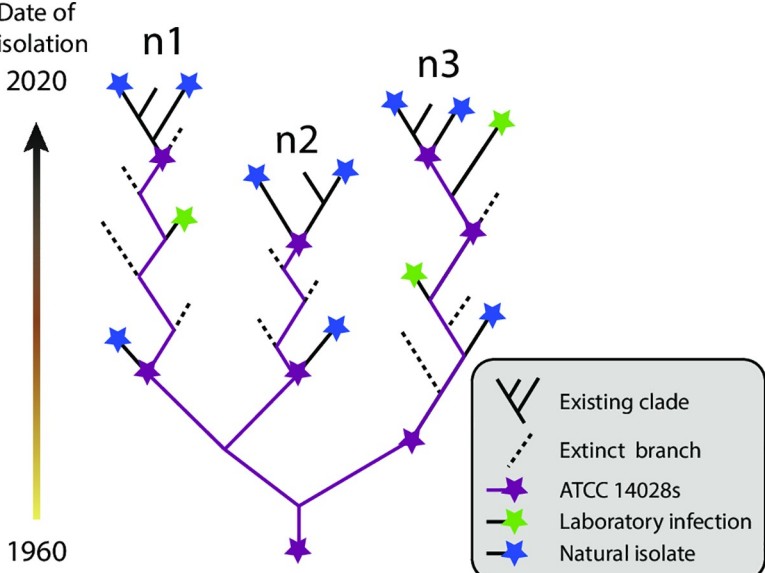

**Fig 7. Cartoon of the microevolution inferred for HC20_373 since the original isolation of ATCC 14028s in 1960.**
Extinct branches from which genomes were not sequenced are indicated by dashed lines whereas extinct branches with genomic sequences are terminated by stars. Colors indicate sources of bacterial isolates.

anthrax [48,49] typhoid fever [50] or SARS [51,52]. Other escapes have caused epidemic or pandemic outbreaks of disease, including H1N1 influenza in 1977 [53] and Venezuelan equine encephalitis [52]. The data presented here add *S. enterica* serovar Typhimurium ATCC 14028s to the list of microbial escapees, with the somewhat exceptional feature that stocks of these bacteria are maintained by multiple commercial providers as well as individual laboratories, and that multiple occasions of escape can be inferred to have happened at the global level. Close relatives of ATCC 14028s have been isolated from rivers (Salinas, U.S.A.; Sinaloa, Mexico) and plants (fruit, vegetables, nuts), and are a continuing source of infections of humans and animals.

## Documented escapes

Extensive epidemiological evidence confirms that ATCC14028s has repeatedly been shed from the laboratory into nature. The first documented escape was in 1984 when ATCC14028s infected at least 750 individuals due to attempted bioterrorism in Oregon [5]. It has caused multiple human cases of gastroenteritis after laboratory infections, as described here and by others [9–13]. Similar to historical cases of shedding by long-term carriers of serovar Typhi [7,8], gastroenteritis will result in the fecal shedding of ATCC 14028s into the sewage system by individuals who were infected in a laboratory. We also identified a surprising, unexpected possible alternative source of release into the environment and/or as a source of human infection. Concern about institutional safety in California, Arizona and France has resulted in draconic regulations which discourage the use of institutional autoclaving for the decontamination of biological waste. Biological decontamination in those areas is now being predominantly performed by commercial companies who collect sealed bags of waste. One of the three human infections with HC20_373 bacteria in France was diagnosed in a truck driver with a mixed infection with two distinct genotypes. This individual reported an incident during the transport of waste from a facility where ATCC14028s is used for quality control, and we anticipate that commercial disposal of bagged biological waste may have released ATCC14028s to the environment on multiple other occasions.

## Rapid extinction of ATCC 14028s in nature

We expected to find natural descendants and sibs of the ATCC14028s derivatives which have repeatedly been shed into nature since 1960. However, similar to prior results with much smaller sets of genomes [9,10], the core genomes of natural isolates of HC20_373 did not demonstrate much additional genetic diversity beyond that found among laboratory strains and laboratory infections (Fig 4). We therefore conclude that longer lasting microevolution within HC20_373 has primarily occurred within the laboratory. We are not aware of descendants of ATCC 14028s that have been associated with extensive transmission chains, and attribute the absence of microevolution in nature to repeated extinction events for natural isolates (Fig 7). Previous results with serovar Paratyphi A also indicated that most Darwinian selection is only transient [54]. The survival time in nature is uncertain. Strains belonging to HC20_373 were isolated from humans over a period of 12 years in Taiwan and 21 years in Ireland, two countries in which ATCC14028s is not used as a quality control for *Salmonella* cultivation and where escape from a laboratory stock might be rare. The BEAST dating tree contains multiple branches of 10-year length or greater. Thus, strains that have escaped from the laboratory might not go extinct for up to 10–20 years after their escape. It seems unlikely that even longer periods of natural expansion are very common because the tMRCA for natural HC20_373 isolates is 1970–1983 and its closest genetic neighbors differentiated from HC20_373 in 1858–1927 (Fig 6). Thus, descendants or sibs of the ATCC14028s that infected chickens in 1960 or

escaped from the laboratory before the 1970s have since disappeared or become exceedingly rare.

## Microevolution of the accessory genome

Nine percent of natural isolates acquired a plasmid or bacteriophage after their escape from the laboratory, predominantly among bacteria isolated from infections of humans, poultry, and animals other than swine. None of their progeny seemed to be particularly fit because none became common. Occasional HGT of plasmids and lysogenic bacteriophages has also been described previously for serovar Agona [55]. In the absence of data to the contrary, it is possibly safest to interpret such occasional HGT as marking the spread of selfish genes rather than the effects of natural selection for particular phenotypes. However, it might be interesting to investigate the frequency of these plasmids within databases of all conjugative plasmids that are currently being established [56]. We have also not attempted a detailed overview of the frequency of the individual bacteriophages detected here but we note that many are of types that are relatively common within *Salmonella*.

## Summary

We describe the natural history of a common laboratory workhorse, ATCC 14028s. This bacterial strain is used by numerous microbiologists as a model organism to study experimental evolution during infections of laboratory animals. It is common used for quality control for *Salmonella* growth in the laboratory, and has been distributed globally across multiple laboratories by post and courier. It is also a common contaminant, not only of culture media but also of laboratory staff, in whom it has caused multiple severe infections. Such contamination events are sufficiently frequent and sufficiently well known that we experienced insurmountable difficulties with a prior attempt to publish a description of natural isolates of HC20_373 because the reviewers claimed that all such "natural isolates" were artefacts resulting from laboratory contamination.

The data presented here are consistent with the interpretation that HC20_373 has escaped from the laboratory on multiple occasions, contaminated the environment and food, and infected humans and other animals around the globe. Genomes from natural isolates of HC20_373 comprised more than 1% of all *Salmonella* genomes in EnteroBase, i.e. these bacteria are more frequently isolated than are *Salmonella* of many rare serovars. It is not possible to confirm that all "natural" isolates were truly natural rather than resulting from laboratory contamination. But the identification of novel plasmids and lysogenic bacteriophages in 9% of natural isolates is a reliable indicator that many or most of them are truly natural. Thus, HC20_373 exists in nature. But it does not seem to continue to transmit indefinitely, and the phylogenetic structure of the natural isolates corresponds to that of repeated extinction events after multiple escapes.

## Methods

### Bacterial isolates

In May 2020, the EnteroBase *Salmonella* database [15] contained >255,000 assembled genomes. We used the Search functions in EnteroBase (https://enterobase.readthedocs.io/en/latest//features/main-search-page.html) to identify all genomes assigned to HC20_373. Two exceptionally divergent draft genomes from laboratory derivatives were excluded from further analysis because the assemblies were of poor quality and others were excluded because their origins were uncertain (S1 Table). The remaining 498 entries were saved as a publicly

accessible workspace (http://enterobase.warwick.ac.uk/a/44709). The metadata were curated manually to ensure that they accurately reflected the original information in ENA. 155 genomes were assigned to the category ATCC 14028s derivative as described in Results. We also assigned 17 genomes to laboratory infections due to publications [9,10] (11 genomes), epidemiological investigations in France (3 genomes; see results) and/or based on public metadata (3 genomes). Other genomes were assigned to the other categories used to color-code Fig 4, and all these assignments were stored in the "14028s" user-defined field within EnteroBase [15]. Genomes assigned to "ATCC14028s derivative" were stored as substrains of Uberstrain SAL_EA9729AA, which has the Name 14028s and corresponds to the complete genome with accession number CP001363.1 which was sequenced by Jarvik *et al.* [1]. On loading this workspace into an internet browser, this Uberstrain is collapsed by default and needs to be expanded by clicking on a triangle symbol in the Uberstrain column.

### Genomic assembly and cgMLST assignments

cgMLST assignments and HierCC clustering are performed automatically on each new genome by EnteroBase [15]. Ten of the 498 genomes in HC20_373 had been downloaded directly from NCBI as complete genomes and the complete genome of NCTC 12023 Colindale (EnteroBase strain barcode SAL_FB4645AA) was assembled from long reads and polished according to short read sequences (S1 Text), whereas the remaining 487 genomes had been assembled by the EBAssembly pipeline V4.1 implemented in EnteroBase [14] from short reads which had been downloaded from NCBI (453 genomes) or uploaded by EnteroBase users (33). The backend pipeline of EnteroBase also automatically generates genomic annotations of all genomes with Prokka against a consistent gene naming scheme, and makes them publicly available for downloading, including all genomes referred to in this publication.

### Core genome phylogeny

The EToKi 'align' module [15] was used to remove repetitive sequence stretches and generate an alignment of the relaxed core genome against the reference genome of 14028s (CP001363.1) for all genomes in HC20_373 plus an outgroup genome from HC20_147 (strain "SAP17-7699"; EnteroBase strain barcode SAL_AB1180AA). The alignment contains 4.73 MB that are shared by at least 95% of the genomes and 462 SNPs. A RAxML v.8.2.4 maximum-likelihood phylogeny of these 462 core SNPs was constructed with the 'EToKi phylo' module [15], and visualized with GrapeTree [57] (Fig 4; public interactive access at http://enterobase.warwick.ac.uk/a/54094).

### Pan-genome construction

A pan-genome of genes of over 300 bp in length was calculated from all HC20_373 genomes using PEPPAN [58] with the parameters '—min_cds 300'. Genes of <300 bp were excluded because they are particular prone to assembly errors. Compatibility of the pan-genome with the reference sequences in the *Salmonella* whole genome MLST scheme was ensured by also specifying the '-g' parameter. The resulting pan genome contains 4627 orthologous groups, of which 4351 were already present in the wgMLST scheme and 276 were novel. One continuous segment of DNA encoding 19 genes in SAL_HC8788AA_AS was scored as contaminating DNA because it was almost identical to a sequence from *Veillonella parvula*, as was a second segment of 5 genes carried by 23 genomes which was identical to sequences in PhiX prophage (S5B Table). These two segments were excluded from further analysis. The final pan-genome consists of 4603 genes (S5A Table).

## Assignments of genes to genomic islands

We reconstructed the presence or absence of all genes in the pan genome for each internal node of a core genome maximum-likelihood phylogeny that had been constructed with Tree-Time [59]. The most recent common ancestor (MRCA) of HC20_373 contained 4107 genes which were interpreted as "ancestral". Deletions of at least five continuous genes in any internal node were scored as large deletions of genomic islands. 496 genes were acquired at internal nodes, and these were assigned to new genomic islands as previously described [60]. In brief, a directed graph was constructed for pairs of orthologous genes that were co-located on a single contig or on pairs of contigs that were linked by read-pairs that straddled both of them. The most likely gene order of the pan genome was identified with Concorde [61] as consisting of the shortest possible path that visited all the genes in the graph. That gene order was manually revised to break and re-join links to duplicated genes and collapsed repeats. All genomic islands are listed in S5D Table, summarized in S6 Table, and illustrated in S2 Fig.

## Temporal signal

The strength of temporal signals in these genomes was evaluated by a Bayesian Evaluation of Temporal Signal (BETS) analysis [39] using BEAST v1.10.5pre [40]. We performed BETS runs for each HC20 cluster in Table 2. For each dataset, we compared relaxed substitution clock and strict substitution clock models with the correct sampling times (heterochronous) and with (artificial) uniform sampling times (isochronous). Marginal likelihoods were generated using generalized stepping-stone sampling (GSS) for each clock model [62], and Log Bayes factors were calculated as the difference between the marginal likelihoods for heterochronous and isochronous data for that model. Models with a Log Bayes factor of 5 or more were accepted as having strong significant evidence for a temporal signal. The clock model with the greater Bayes Factor was used to estimate dated phylogenies from the heterochronous data with statistical temporal support using the HKY+G4 substitution model and tree priors estimated according to an exponential growth model. For each HC20 cluster, we ran three independent Markov chain Monte Carlo analyses for 100 million generations, and combined 1,800 samples post burn-in from each chain to compute posterior summaries (posterior medians and 95% highest posterior density intervals).

## Supporting information

**S1 Text. Genomic island instability in HC20_373.** History of ATCC 14028s. History of CIP 104115. History of NCTC 12023. History of NCTC12023 David Holden. History of NCTC12023 NalR Hensel. History of NCTC12023 Gerlach. Random large HC20 clusters. Outbreaks associated with HC20 clusters. HC20 clusters of laboratory strains. NCTC 7832. HC20_5519.
(DOCX)

**S1 Table. Reassigned categories for genomes whose supposed source or date of isolation according to public metadata was suspect plus details on outgroup genome.**
(XLSX)

**S2 Table. Dated depositions of ATCC14028s derivatives in multiple laboratories, including node assignments and distinctive SNPs.**
(DOCX)

**S3 Table. Metadata for 496 genomes of HC20_373 from diverse sources plus one outgroup genome from HC20_147.**
(XLSX)

**S4 Table. Properties of additional HC20 Clusters with large cluster numbers with low HC_20 numbers (25.06.2021).**
(XLSX)

**S5 Table S5A Table A pan-genome of 4603 genes with ≥300 bp length, and its distribution across 496 HC20_373 genomes S5B Table Pan-genome genes which were deleted because they reflected contamination S5C Table Genomics islands present or absent in individual genomes from 496 entries S5D Table 98 events of gain or loss of genomic islands in individual genomes.**
(XLSX)

**S6 Table. Summary of larger InDels observed in HC20_373 genomes.**
(DOCX)

**S7 Table. Molecular clock rates in other *S. enterica* clades.**
(DOCX)

**S1 Fig. Log-log plot of frequency of *Salmonella* HC20 clusters by numbers of genomes in each cluster.** This figure is an alternative representation of the data summarized in Fig 5A.
(PDF)

**S2 Fig. Powerpoint summary of genetic content of genomic islands that were lost and gained within HC20_373.**
(PPTX)

**S3 Fig. Bayesian BEAST [40] temporal dating of ATCC14028s and its natural derivatives.** As Fig 6 except that EnteroBase genome barcodes are shown in miniscule fonts at the right of the tree.
(PDF)

**S4 Fig.** Ninja NJ visualization of allelic differences in the 3002 core genes of the cgMLST *Salmonella* scheme with GrapeTree for genomes within A) HC20_10 and B) HC20_44. Further information on these HC20 clusters is summarized in Tables 2, 4 and 5, and an interactive version of both trees can be accessed at https://enterobase.warwick.ac.uk/ms_tree?tree_id=45443 and https://enterobase.warwick.ac.uk/ms_tree?tree_id=45449.
(PDF)

**S5 Fig.** Ninja NJ visualization of allelic differences in the 3002 core genes of the cgMLST *Salmonella* scheme with GrapeTree for genomes within A) HC20_122 and B) HC20_125. Further information on these HC20 clusters is summarized in Tables 2, 4 and 5, and an interactive version of both trees can be accessed at https://enterobase.warwick.ac.uk/ms_tree?tree_id=45447 and https://enterobase.warwick.ac.uk/ms_tree?tree_id=45450.
(PDF)

**S6 Fig.** Ninja NJ visualization of allelic differences in the 3002 core genes of the cgMLST *Salmonella* scheme with GrapeTree for genomes within A) HC20_557 and B) HC20_710. Further information on these HC20 clusters is summarized in Tables 2, 4 and 5, and an interactive version of both trees can be accessed at https://enterobase.warwick.ac.uk/ms_tree?tree_id=45446 and https://enterobase.warwick.ac.uk/ms_tree?tree_id=45445.
(PDF)

**S7 Fig.** Ninja NJ visualization of allelic differences in the 3002 core genes of the cgMLST *Salmonella* scheme with GrapeTree for genomes within A) HC20_1348 and B) HC20_1487. Further information on these HC20 clusters is summarized in Tables 2, 4 and 5, and an interactive version of both trees can be accessed at https://enterobase.warwick.ac.uk/ms_tree?tree_id=45423 and https://enterobase.warwick.ac.uk/ms_tree?tree_id=45424.
(PDF)

**S8 Fig.** Ninja NJ visualization of allelic differences in the 3002 core genes of the cgMLST *Salmonella* scheme with GrapeTree for genomes within A) HC20_39803 and B) HC20_4179. Further information on these HC20 clusters is summarized in Tables 2, 4 and 5, and an interactive version of both trees can be accessed at https://enterobase.warwick.ac.uk/ms_tree?tree_id=45421 and https://enterobase.warwick.ac.uk/ms_tree?tree_id=60146.
(PDF)

**S9 Fig.** Ninja NJ visualization of allelic differences in the 3002 core genes of the cgMLST *Salmonella* scheme with GrapeTree for genomes within A) HC20_20633 and B) HC20_5519. Further information on these HC20 clusters is summarized in Tables 2, 4 and 5, and an interactive version of both trees can be accessed at https://enterobase.warwick.ac.uk/ms_tree?tree_id=60214 and https://enterobase.warwick.ac.uk/ms_tree?tree_id=60272.
(PDF)

## Acknowledgments

The ATCC14028s Study Group consisted of Steven Huynh[a], Lisa Gorski[a], Anita S. Liang[a], Ohannes Guerbidjian[f], Chien-Shun Chiou[g], Niall Delappe[h], Heike Claus[i], François-Xavier Weill[j], Derek Brown[k], Eva Litrup[l] and Mia Torpdahl[l,] Anthony M. Smith[m] and Jake David Turnball[n]. We also gratefully acknowledge the receipt of bacterial strains and/or DNA as well as historical information from David Holden, Michael Hensel and Roman Gerlich.

Addresses of the members of the ATCC14028s Study Group: [a]USDA, Agricultural Research Service, Albany, CA 94710, U.S.A., [f]Dept. of Biology, California State University—Northridge, Northridge, CA 91330, U.S.A., [g]Central Regional Laboratory, Center for Diagnostics and Vaccine Development, Centers for Disease Control, Taichung, Taiwan, [h]National Salmonella, Shigella and Listeria Reference Laboratory, Galway, Ireland, [i]Institut für Hygiene und Mikrobiologie, Universität Würzburg, 97080 Würzburg, Germany, [j]Institut Pasteur, Unité des Bactéries pathogènes entériques, 75724 Paris cedex 15, France, [k]Scottish Microbiology Reference Laboratory, Glasgow, UK, [l]SSI, Copenhagen, Denmark, [m]Centre for Enteric Diseases, National Institute for Communicable Diseases, Johannesburg, Gauteng 2131, South Africa, [n]The National Collection of Type Cultures, Colindale, London, UK.

## Author Contributions

**Conceptualization:** Mark Achtman, Philippe Lemey, Craig T. Parker.

**Data curation:** Mark Achtman, Kerry K. Cooper, Craig T. Parker, Zhemin Zhou.

**Formal analysis:** Mark Achtman, Frederik Van den Broeck, Philippe Lemey, Zhemin Zhou.

**Funding acquisition:** Mark Achtman, Philippe Lemey, Craig T. Parker.

**Investigation:** Mark Achtman, Frederik Van den Broeck, Kerry K. Cooper, Craig T. Parker, Zhemin Zhou.

**Methodology:** Frederik Van den Broeck, Philippe Lemey, Zhemin Zhou.

**Project administration:** Mark Achtman, Philippe Lemey, Craig T. Parker.

**Resources:** Craig T. Parker, Zhemin Zhou.

**Software:** Philippe Lemey, Zhemin Zhou.

**Supervision:** Mark Achtman, Philippe Lemey, Craig T. Parker.

**Validation:** Mark Achtman, Zhemin Zhou.

**Visualization:** Mark Achtman, Frederik Van den Broeck, Zhemin Zhou.

**Writing – original draft:** Mark Achtman, Frederik Van den Broeck.

**Writing – review & editing:** Mark Achtman, Kerry K. Cooper, Philippe Lemey, Craig T. Parker, Zhemin Zhou.

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
