## [Decision Letter · Decision Letter 0]

10 Jun 2021

Dear Dr Achtman,

Thank you very much for submitting your Research Article entitled 'Repeated transient escape of a laboratory strain of Salmonella enterica into nature' to PLOS Genetics.

The manuscript was fully evaluated at the editorial level and by independent peer reviewers. The reviewers appreciated the attention to an important problem, but raised some substantial concerns about the current manuscript. A key point was that the claim of repeated  "lab escapes" was derived from just one interpretation of the population structure, and lacked discussion of alternative hypotheses.  Based on the reviews, we will not be able to accept this version of the manuscript, but we would be willing to review a much-revised version. We cannot, of course, promise publication at that time.

If you decide to revise the manuscript for further consideration at PLOS Genetics, please aim to resubmit within the next 60 days, unless it will take extra time to address the concerns of the reviewers, in which case we would appreciate an expected resubmission date by email to plosgenetics@plos.org.

[LINK]

We are sorry that we cannot be more positive about your manuscript at this stage. Please do not hesitate to contact us if you have any concerns or questions.

Yours sincerely,

Jay C. D. Hinton

Guest Editor

PLOS Genetics

Hua Tang

Section Editor: Natural Variation

PLOS Genetics

Reviewer's Responses to Questions

**Comments to the Authors:**

Reviewer #1: The manuscript by Achtman et al builds upon their previous sophisticated genome analysis studies. Here, they are testing the hypothesis that the laboratory strain ATCC14028s is widespread in the environment and therefore regularly contaminates food products. The authors compared the properties of >280,000 Salmonella genomes in EnteroBase, including laboratory and diverse natural bacterial strains with ATCC14028s-like genomes. The natural isolates within HC20_373 are distinct from other natural isolates, cluster tightly, and appear to be descendants of ATCC14028s caused by laboratory contamination. The authors conclude that ~8% of natural isolates acquired mobile genetic elements by horizontal gene transfer and that ATCC14028s does not survive long-term in the environment.

Major points

This manuscript will be of interest to people who work on Salmonella enterica and to lab safety microbiologists. However, if you take a step back, it basically says that horizontal gene transfer occurs in the wild and that a lab strain cannot survive in the wild. These ideas are already well-established.

Minor points

The manuscript is highly repetitive.

Reviewer #2: Achtman et al., have written a clear and interesting manuscript describing the repeated release of a strain of S. Typhimurium into the environment, resulting in human infections and environmental contamination. There are many fascinating titbits in this manuscript — I particularly enjoyed the tracing of one sub-clade of ATCC 14028s across London, and over to Germany. It was very interesting that one culture at Colindale held three minor variants which were observed in different sub-clades, reminiscent of the AmeriThrax investigation. I was re-assured of the veracity of the authors findings by the fact that it was observed causing infections by labs which don’t use it as a control, because otherwise, there is always the chance that it was due to contamination of the patient sample/culture by the control culture.

As might be expected from this group of authors, the results presented are technically very solid and the interpretations entirely reasonable. I particularly enjoyed this interesting illustration of the fact that most extant bacterial lineages go extinct, not because of any difference in fitness, but just through stochastic processes. Many researchers go the other way and try to find genetic variations which explain every observation, so this is most welcome.

Overall, this is a unique and very interesting manuscript of a kind of “natural experiment”, and I very much enjoyed the manuscript.

I have a few minor comments:

1. It is fascinating that the epidemiological properties of HC_20_373 differ from that of the other HC20s in its HC50. The difference in geography makes sense because ATCC 14028s has had a major boost through the international postal service, but what explains the different proportion from non-human sources? Sampling bias?

2. What exactly do you mean when you use the term “founder nodes”?

3. Why was the tMRCA of the HC20 20 years after the isolation of ATCC 14028s?

4. P18 – “Furthermore, the MCC tree root of HC20_373 was fixed at node C1 in the ML tree, which is the node at which the branch from an outgroup genome in HC20_147 joins the HC20_373 cluster (Fig. 4)”. This seems discrepant with what is depicted in Figure 4. Quoting from Figure 4 legend – “the phylogenetic root of HC20_373 is indicated by the branch connecting node A1 with the outgroup.”

5. P20 - "Furthermore, many microbiology laboratories first streak a bacterial culture to single colonies before storage, which constitutes an absolute bottleneck that will immediately fix any rare mutations that happen to have segregated to that single colony" but there is limited evidence of this effect, because the clock rate is slower than other S. enterica.

And some very minor:

6. p6 - “which might be a more cause of parallel outbreaks at the national level”. Please rephrase, missing word perhaps?

7. What are the percentages on Figure 4

Reviewer #3: Please see attached review

Reviewer #4: Achtman et al describe a straightforward comparative genomic and phylogenetic analysis of S. Typhimurium ATCC14028 and related strains present in their Enterobase database. They report that natural isolates are very closely related to multiple sequences of ATCC14028 from laboratories and that together lab and natural isolates form a genetically isolated group. The authors conclude therefore that the natural isolates are due to repeated escape of laboratory strains. The authors suggest that a small proportion of the natural isolates contain plasmids or prophage that distinguish them from laboratory strains, suggests at least limited growth and transmission within animals outside of the lab. This is a technically excellent study and a well-presented manuscript. However, the conclusion that HC20-737 has repeatedly escaped from labs into nature is not well supported by the data or particularly plausible. The observation that this HC has very limited diversity and multiple short chains link laboratory and natural isolates is perplexing, but that this is completely atypical or indeed a common feature of lab-escape strains is not addressed. It is clear that one possible explanation is escape from the labs, but some equally compelling counterarguments should be addressed. One of these is the curious lack of evidence that natural isolates have been genetically manipulated, something that is very common in research laboratories. Also, the implausibility that such high numbers of natural infections can be sustained by a strain that is only transiently present outside of the lab and therefore does not have time to circulate widely in zoonotic sources. Rather than directly test their hypothesis and examine its plausibility, the authors provide sweeping statements such as the presence of phage and plasmids in ‘natural’ isolates is evidence of having circulated in animal populations, ignoring the possibility that these have been acquired by HGT in the lab.

General comments

1. As a consideration to the reviewer, it would be appreciated if line numbers could be included in submitted manuscript, in line with the journals instructions.

2. Was there a difference in the proportion of animal and human HC20-737 isolates that had gained plasmids or prophage? It might be expected that considering the transient nature and limited distribution of hypothetically escaped 14028 strains in animals that many of the human isolates would have been from unreported lab contamination and therefore be less likely to have circulated in the environment for an extended period of time to acquire these elements. That animal isolates will not have been lab acquired infections these would be expected to have an elevated level of HGT.

3. How atypical is the diversity of HC20-737? Some analysis of the variation in diversity at the HC20 level would be useful for perspective and may identify other examples of frequent laboratory escape. The case that HC20-737 is less diverse than other HC50_147 clusters, but are there other examples of similarly low diversity HC20 clusters? Analysis of clonal groups that are and others that are not widely available from ATCC or used in research labs for historic reasons. Also, an analysis of possible lab escape of the recommended strain for quality control in the UK ATCC7832 (serovar Nottingham) as this is the country where much of the natural isolates were sequenced since 2014 originated. Is it possible to identify the rate at which lab cross-contamination occurs from the example of ATCC7832 (serovar Nottingham)? It seems this is likely to be a rare serotype and may be why it was chosen for quality control, and laboratory escape should be easy to identify.

4. It is reported that some lab strains of 14028 contained transposons used in genetic manipulations and this was not found in any HC20-737 natural isolates. It is not clear how this is consistent with the conclusion that natural isolates are from frequent contamination from lab escape, since if this was the case we would expect these transposons to be present in at least some of the natural isolates. Why is there no evidence of laboratory manipulation of natural isolates if they have come from research labs?

5. Most of the laboratory isolates that are sequenced are likely to be the wild type master strain. It is extremely common to genetically modify lab strains by allelic exchange and these are obvious examples of genetic manipulation. Commonly these strains are actually designed to be unlikely to be attenuated and these should also be escaping from labs at a high rate along with the wildtype parent strain. Why is there no evidence of these in natural isolates? For nearly 2 decades now mutations have been introduced by recombineering methods that involve insertion of antibiotic resistance genes and leave FLP-site scars in the genome if resolved, is there any evidence of these manipulations?

6. How many labs in the UK routinely culture strain 14028? Since the authors establish that lab escape is unlikely to be from public health labs in the UK as this strain is not used for quality control, presumably the authors suspect research labs, or perhaps industry? Please clarify. The number of research labs using 14028 could be estimated from searching literature reporting experiments labs during the period 2016-2020 when most of the experiments are likely to have been performed that could have resulted in lab escape in the period covered by much of the sequencing of natural isolates in the UK from 2014 onwards. Other commonly used lab strains used in this period would also be expected to appear as natural isolates if lab escape is as frequent as suggested by the conclusions of this manuscript. Is there evidence for the escape of strain SL1344, for example?

7. It is proposed that between 2014 and 2020, 74 cases of Salmonellosis in the United Kingdom were caused by HC20-737 that had escaped from labs, 0.14% of cases for which sequence data were available. In Ireland (Rep.) there were 14 cases, 0.7% of genomes. Considering that the hypothetically escaped HC20-737 do not appear particularly fit since they never establish themselves in animal populations as evidenced by the lack of diversity, the plausibility of the conclusions drawn by the authors should be addressed. In particular, the positivity rates of livestock and poultry (pigs around 20%) that represent host populations where it is known that some strains of Salmonella have been circulating for decades and are likely to represent a considerably larger pool of Salmonella capable of causing human infections. This is especially true since a short-lived escape strain is likely to have little opportunity to spread widely in zoonotic host species, limiting the likelihood that it can contaminate a sizable proportion of the food consumed. What frequency of lab escape is likely to be needed in order to support a natural infection rate that leads to 0.7% of infections being of the escaped type?

8. The significance of different frequency of SPI deletion in lab strains and natural isolates to the research question is not clear.

9. A technical description of GSS analysis that was used to test for a temporal signal in the laboratory and natural isolates of HC20-737 is included but is likely to be impenetrable to the non-expert and at least some interpretation should be included in the discussion. It would be expected that this analysis is unlikely to be informative due to the limited sequence variation and the fact that accumulation of SNPs is likely to vary based on the frequency of sub-culture and microbiology practices such as method of storage and subculture. The rationale for doing this analysis is therefore not clear and no conclusions are drawn. This section should therefore be removed.

Specific comments

1. abstract, It is not clear to which 'Epidemiological features indicate that the natural isolates do not represent recent contamination

by the laboratory strain' are being referred?

2. p13-14. The section 'Are “natural” isolates truly natural?' does not convincingly make the argument that the majority or indeed any of the proposed natural isolates are indeed natural. The first paragraph seems to make the argument that because the authors’ manual curation identified some cases where supposed natural isolates were in fact not natural is an argument that they have identified all of these. If this was not the argument then please clarify. That an isolate from a baby with no known contacts to a laboratory or waste does not discount this possibility that this did not occur. The lack of evidence is not good evidence in this instance. The second paragraph does contain more persuasive arguments against laboratory cross-contamination and suggests that the proposed lab-escape in the UK is not from public health labs but rather research labs or industry. If this is the conclusion then it should be clearly stated as it has some important considerations about the nature of the strains that escape such as evidence of genetic engineering.

3. p21. Please provide evidence that Salmonella can ‘establish itself in rivers and on plants’. Perhaps I have misunderstood the intention of this statement, but Salmonella is not generally considered an environmental microorganism and when found there is generally considered to be there transiently as a result of fecal contamination that while is likely to be important for transmission is not thought to represent a mode of living for extended periods of time. More relevant to the study is whether escaped 14028 has established itself in animal populations.

4. p21. ‘infected the environment’ consider editing to ‘contaminated the environment’ and ‘environmental environments’ seems redundant use.

5. Summary. The description of prior attempts to publish the work and the reviewers objections are not relevant and should be removed from the manuscript.

6. The conclusion that the presence of phage and plasmids in proposed natural HC20-737 strains indicates that they circulated in animals outside of the lab is not well supported. Please provide evidence to support this kind of statement. Both public health and research labs culture many different bacteria and horizontal gene transfer in the laboratory is absolutely possible considering previous statements indicating the frequency of lab contamination and infection of personnel.

**Have all data underlying the figures and results presented in the manuscript been provided?**

Reviewer #1: Yes

Reviewer #2: Yes

Reviewer #3: Yes

Reviewer #4: Yes

PLOS authors have the option to publish the peer review history of their article (what does this mean?). If published, this will include your full peer review and any attached files.

Reviewer #1: No

Reviewer #2: No

Reviewer #3: **Yes: **Dr Marie Anne Chattaway

Reviewer #4: No

---

## [Decision Letter · Decision Letter 1]

26 Aug 2021

Dear Dr Achtman,

Thank you very much for submitting your Research Article entitled 'Repeated transient escape of a laboratory strain of Salmonella enterica into nature' to PLOS Genetics.

The manuscript was fully evaluated at the editorial level and by independent peer reviewers. The reviewers appreciated the attention to an important problem, but raised some substantial concerns about the current manuscript. Based on the reviews, we will not be able to accept this version of the manuscript, but we would be willing to review a much-revised version that addresses the comments of Reviewer #4. We cannot, of course, promise publication at that time.

Please see the comments below from the Guest Editor which several senior editors, including the Editors-in-Chief, agree with. We will not be able to move forward, unless a revised version of the manuscript can rigorously discuss alternative interpretations.

If you decide to revise the manuscript for further consideration at PLOS Genetics, please aim to resubmit within the next 60 days, unless it will take extra time to address the concerns of the reviewers, in which case we would appreciate an expected resubmission date by email to plosgenetics@plos.org.

[LINK]

We are sorry that we cannot be more positive about your manuscript at this stage. Please do not hesitate to contact us if you have any concerns or questions.

Yours sincerely,

Jay C. D. Hinton

Guest Editor

PLOS Genetics

Hua Tang

Section Editor: Natural Variation

PLOS Genetics

Editor's Comments

**Comments to the Authors:**

Your 14028 paper is a tour-de-force that shows the remarkable level of genome-derived detail that can contribute to the analysis of an important cluster of S. Typhimurium strains. You report an intricate detective story that will be of great interest to the microbiological community.

However, the consensus of the reviewers is that a key limitation of the current manuscript is that it only presents one interpretation of the data: that the group of genomes represented by hierarchical cluster HC20_373 are largely derived from S. Typhimurium 14028s strains that have escaped from the laboratory. Reviewer #4 states that you have not unequivocally demonstrated that the data indicates repeated transient escape of the laboratory strain.

We consider that the best way forward for your manuscript is for many of your findings to be expressed more equivocally. For example, at line 551 – 553, it is stated that S. Typhimurium isolates from rivers and plants are “descendants of ATCC14028s”. Is it not possible that these isolates are simply close relatives of CDC 60-6516, and have persisted in the environment for decades?

At line 537 – 540, it is stated that the “observations indicate that the natural isolates within HC20_373 represent the descendants of repeated escape from the laboratory”. It would be safer to state that the observations would be “consistent with” such an interpretation. The new information in Table 2 clearly shows that fewer pairwise SNP differences distinguish isolates belonging to the HC20_373 than other HC20 clusters. Another interpretation of this finding is that the homogeneity of HC20_373 could reflect a lack of evolutionary pressure that act on the bacteria in certain ecological niches.

At line 620, it is stated that “The data presented here show that has escaped from the laboratory on multiple occasions…”. Again, the data are consistent with such an interpretation - but the claim remains unproven.

One of the reasons that ATCC14028s has been so widely adopted by bacterial researchers and diagnostic laboratories is that it was derived from the chicken isolate CDC 60-6516, and so represents a “natural isolate”. Because ATCC14028s remains virulent in many animal models, perhaps it should not be described as a “laboratory strain”in the title of the paper. Rather, ATCC14028s is a natural isolate of S. Typhimurium that has been studied for decades in the laboratory.

As suggested by Reviewer #4, your “analysis needs to be interpreted more cautiously, and presented in a more balanced way”. Please modify the title of your manuscript to reflect the fact that “repeated transient escape” is one possible interpretation of your data.

Please revise your manuscript to address the comments of Reviewer #4.

Reviewer's Responses to Questions

**Comments to the Authors:**

Reviewer #1: The manuscript is significantly improved with regard to writing. However, it still “basically says that horizontal gene transfer occurs in the wild and that a lab strain cannot survive in the wild. These ideas are already well-established.” The authors “agree that the idea that HGT happens in Nature has been widely accepted” but state that “the temporal dynamics of this process have been adequately defined”. However, the details of the length of survival of escaped organisms are likely to vary depending on the biology of the organism and the micro/macro environments into which they were released, indicating that they belong in specialty journals unless they reveal larger truths.

Re the second point, that it is well established that lab strains cannot survive in the wild, instead of addressing the criticism, the authors quibble over the meaning of “survival” versus “persistence”. They write, “The claim that it is well established that a lab strain cannot survive in the wild is refuted by this paper. ATCC14028s survives quite well… What is being addressed here is whether it (ATCC14028s ) will persist”. Nevertheless, the manuscript concludes that “the release of ATCC14028s in the environment is not accompanied by long-term persistence and additional microevolution”, confirming that there is no new idea here.

Reviewer #2: No further comments.

Reviewer #4: It remains my opinion that the authors have not unequivocally demonstrated that the data indicates repeated transient escape of a laboratory strain. It is also my opinion that the data suggests that this may happen but great caution with this conclusion is warranted. This is at the heart of the problem I have with the manuscript in its current form: it is written with an unequivocal conclusion that can in fact only ever be equivocal. The title is wholly inappropriate for the level of certainty, and the text also presents this conclusion in a similar tone.

The level of effort made by the authors to address my concerns particularly through presentation of additional analysis and description in supplementary material is appreciated. The authors state that this information was not previously included because ‘we were concerned that bringing in additional data would make it more difficult to make our points’. This appears to be the case since the additional data and analysis does not add support for the conclusions. The analysis needs to be interpreted more cautiously and presented in a more balanced way acknowledging that the authors cannot be certain of their main conclusion. This problem is most apparent in the title but is the case throughout the results and discussion and even the introduction. The first sentence of the introduction is a circular argument where the question posed ‘How extensively do Salmonella diversify when they escape from the laboratory into nature’ entirely relies on the answer arrived at in the study. The additional analysis did not lead to the identification of any other examples of potential lab escape, with the possible exception of S. Nottingham, and it was conceded that the low genetic diversity of these isolates and geographical distribution may be entirely due to lab cross-contamination. Another example of low genetic diversity was the CC20 containing lab strain SL1344 and it was concluded that there was little evidence of lab escape for this strain. HC20-27280 is the most similar in diversity to HC20-373 but does not appear to be mentioned anywhere. We are therefore left to assume that 14028 is a special case but are not offered any compelling evidence as to why. If lab escape is so common, why is it only evident for 14028?

The authors dismiss my comment about the possibility of transfer of mobile genetic elements in the lab. I stand by my assertion that mobile genetic elements can and do get transferred in the lab where diverse bacteria are cultured, although I do concede that this is not reported in the literature. It happens at increased frequency when sharing a lab with researchers working on phage. Acquisition of mobile elements does not require co-culture and this was not mentioned in my comments. This can and does happen through contaminated glassware / pipettes etc. This is probably more likely to happen in public health labs where numerous different species are cultured and mobile genetic elements are likely to be a common contaminant in the lab environment. Indeed, pre-enrichment in non-selective media is the first step in isolation of Salmonella from samples. Evidence of transfer may be less likely in research laboratories where it is common practice to ensure that mutations are present in a ‘clean’ genetic background for example by transduction of markers using P22. P22 lysogens are specifically excluded from analysis by testing for P22 susceptibility. The presence of diverse mobile genetic elements in genome sequence is therefore not a good indication of transient existence in the environment as opposed to acquisition due to cross-contamination in the lab.

The authors express their initial concern at the lack of evidence of genetic manipulation in ‘natural isolates’, but point out that ‘a few did’ escape. It is not clear where this is discussed in the text. On lines 437-438 it is stated that ‘none (<0.3%) of the 324 natural strains from 438 HC20_373 contained a laboratory transposon’.

The authors address the implausibility of such high numbers of natural isolates being sustained by a strain that is transiently present outside of the lab by claiming an assumption was made that transient refers to days. The comments made in my first review did not mention any such assumption. I would extend this to months or even years and come to the same conclusion. When new epidemic strains emerge, the common ancestor existed long before significant numbers of the strain are isolated during surveillance eg PMID 26944846 and 34093465, suggesting that the strain was circulating in low numbers for many years before it was isolated in even low numbers from animals and humans. It does not require detailed modelling to at least have some degree of scepticism about the plausibility of the conclusions drawn in this study.

**Have all data underlying the figures and results presented in the manuscript been provided?**

Reviewer #1: Yes

Reviewer #2: Yes

Reviewer #4: Yes

PLOS authors have the option to publish the peer review history of their article (what does this mean?). If published, this will include your full peer review and any attached files.

Reviewer #1: No

Reviewer #2: No

Reviewer #4: No

---

## [Editor Report · Decision Letter 2]

10 Sep 2021

Dear Dr Achtman,

We are pleased to inform you that your manuscript entitled "Genomic population structure associated with repeated escape of Salmonella enterica ATCC14028s from the laboratory into nature" has been editorially accepted for publication in PLOS Genetics. Congratulations!

Yours sincerely,

Jay C. D. Hinton

Guest Editor

PLOS Genetics

Hua Tang

Section Editor: Natural Variation

PLOS Genetics

Comments from the reviewers (if applicable):

**Data Deposition**

http://datadryad.org/submit?journalID=pgenetics&manu=PGENETICS-D-21-00461R2

**Press Queries**

---

## [Editor Report · Acceptance letter]

22 Sep 2021

PGENETICS-D-21-00461R2 

Genomic population structure associated with repeated escape of Salmonella enterica ATCC14028s from the laboratory into nature 

Dear Dr Achtman, 

We are pleased to inform you that your manuscript entitled "Genomic population structure associated with repeated escape of Salmonella enterica ATCC14028s from the laboratory into nature" has been formally accepted for publication in PLOS Genetics! Your manuscript is now with our production department and you will be notified of the publication date in due course.

With kind regards,

Andrea Szabo

PLOS Genetics

On behalf of:
